# Cell-based HTS identifies a chemical chaperone for preventing ER protein aggregation and proteotoxicity

Keisuke Kitakaze[1,2,3], Shusuke Taniuchi[1,2,4], Eri Kawano[1], Yoshimasa Hamada[1,2], Masato Miyake[1,2,4], Miho Oyadomari[1], Hirotatsu Kojima[5], Hidetaka Kosako[2], Tomoko Kuribara[6], Suguru Yoshida[6], Takamitsu Hosoya[6], Seiichi Oyadomari[1,2,4]*

[1]Division of Molecular Biology, Institute for Genome Research, Institute of Advanced Medical Sciences, Tokushima University, Tokushima, Japan; [2]Fujii Memorial Institute of Medical Sciences, Institute of Advanced Medical Sciences, Tokushima University, Tokushima, Japan; [3]Department of Pharmacology, Kawasaki Medical School, Kurashiki, Japan; [4]Department of Molecular Research, Diabetes Therapeutics and Research Center, Institute of Advanced Medical Sciences, Tokushima University, Tokushima, Japan; [5]Drug Discovery Initiative (DDI), The University of Tokyo, Tokyo, Japan; [6]Laboratory of Chemical Bioscience, Institute of Biomaterials and Bioengineering, Tokyo Medical and Dental University (TMDU), Tokyo, Japan

**Abstract** The endoplasmic reticulum (ER) is responsible for folding secretory and membrane proteins, but disturbed ER proteostasis may lead to protein aggregation and subsequent cellular and clinical pathologies. Chemical chaperones have recently emerged as a potential therapeutic approach for ER stress-related diseases. Here, we identified 2-phenylimidazo[2,1-*b*]benzothiazole derivatives (IBTs) as chemical chaperones in a cell-based high-throughput screen. Biochemical and chemical biology approaches revealed that IBT21 directly binds to unfolded or misfolded proteins and inhibits protein aggregation. Finally, IBT21 prevented cell death caused by chemically induced ER stress and by a proteotoxin, an aggression-prone prion protein. Taken together, our data show the promise of IBTs as potent chemical chaperones that can ameliorate diseases resulting from protein aggregation under ER stress.

*For correspondence:
oyadomar@tokushima-u.ac.jp

**Competing interests:** The authors declare that no competing interests exist.

## Introduction

Proteins are responsible for many tasks in cellular life, and having a correct three-dimensional protein structure is essential for their function. Molecular chaperones are proteins that assist the covalent folding or unfolding of proteins and the assembly or disassembly of protein complexes. One major function of chaperones is to prevent both newly synthesized polypeptide chains and assembled subunits from aggregating into non-functional structures. The accumulation of unfolded or misfolded proteins in the endoplasmic reticulum (ER) is called ER stress. The cell has an adaptive system against ER stress called the unfolded protein response (UPR), which is the coordinated transcriptional upregulation of ER chaperones and folding enzymes that prevents the aggregation of unfolded and incompletely folded proteins (*Mori, 2000*; *Walter and Ron, 2011*; *Han and Kaufman, 2017*). The UPR is initiated by the activation of the following three signal transducers located in the ER: IRE1α, PERK and ATF6. Disturbance of the UPR has important pathological consequences, including diabetes, neurodegenerative disease and cancer (*Hotamisligil, 2010*; *Martínez et al., 2018*; *Clarke et al., 2014*).

To overcome ER stress-related diseases, the following two pharmacological strategies can be applied: the modulation of ER protein folding environments (by UPR modulators) and a reduction in the accumulation of unfolded or misfolded ER proteins (by chemical chaperones). With the past decade of research, encouraging progress has been made in this area through the identification of several compounds. Several UPR modulators that show therapeutic benefits in mouse disease models have been developed. KIRA6, an inhibitor of the UPR regulator IRE1, and azoramide, an ER calcium modulator, have shown potent anti-diabetic efficacy (*Ghosh et al., 2014*; *Fu et al., 2015*). Both Sephin1, a PPP1R15A inhibitor, and ISRIB, an eIF2B activator, modulate PERK signalling and have been reported to exert neuroprotective effects in mouse models of amyotrophic lateral sclerosis (ALS) and prion disease (*Das et al., 2015*; *Halliday et al., 2015*). In contrast, several chemical chaperones that have shown therapeutic benefits in mouse disease models have also been developed. 4-Phenylbutyrate (4PBA) and taurourso-deoxycholic acid (TUDCA) have been reported to function as chemical chaperones and have shown therapeutic benefits for a wide variety of diseases, such as diabetes, ALS and Alzheimer's disease (*Ozcan et al., 2006*; *Cudkowicz et al., 2009*; *Nunes et al., 2012*). These data indicate that pharmacological intervention to alleviate ER stress is a promising approach for treating a range of diseases.

Although chemical chaperones represent an interesting potential novel class of therapeutics, due to limitations including high concentration treatment of currently known chemical chaperones, identifying novel chemical classes of chemical chaperones is desirable. Several approaches for the high-throughput screening (HTS) of small molecules have been used. First, a cell survival-based HTS approach to identify small molecules that protect cells against ER stress-induced cell death was attempted. Indeed, salubrinal (*Boyce et al., 2005*) and compound 13d (*Duan et al., 2017*) were identified by HTS as small molecules to protect neuronal cells or pancreatic β-cells against ER stress-induced apoptosis, respectively. However, this phenotypic HTS has challenges in compound target identification. A molecular target-based approach for drug discovery was also attempted. For example, KIRAs (*Wang et al., 2012*), the PERK inhibitor GSK2656157 (*Axten et al., 2012*), and the IRE1/PERK activator IPA (*Mendez et al., 2015*) were identified or developed from an initial screening hit by structure-based design. Undoubtedly, molecular target-based screening has some advantages over phenotypic screenings, as a molecular target assay is often required for subsequent chemical optimization of lead compounds from HTS hit compounds. However, target-based drug discovery may have limitations because multiple targets and signalling pathways can be involved in pathological conditions. Having the best features of phenotypic and molecular target-based HTS, signalling pathway HTS has gained momentum for identifying UPR modulators. For instance, the PERK inhibitors GSK2606414 (*Axten et al., 2012*) and ISRIB (*Sidrauski et al., 2013*), the IRE1 inhibitors STF-083010 (*Papandreou et al., 2011*) and 4μ8c (*Cross et al., 2012*), the ATF6 inhibitor Ceapins (*Gallagher and Walter, 2016*) and the ATF6 activator AA147 (*Plate et al., 2016*) were identified using PERK, IRE1 and ATF6 reporter assays, respectively.

One of the key obstacles of HTS for chemical chaperones is the lack of a quantitative assay to directly monitor ER protein folding. We attempted to tackle this obstacle by utilizing a UPR signalling assay. Here, we employed a cell-based HTS with a highly sensitive UPR signalling reporter and discovered 2-phenylimidazo[2,1-*b*]benzothiazole derivatives (IBTs) as chemical chaperones. Biochemical and chemical biology approaches revealed that IBT21 directly binds unfolded or misfolded proteins to inhibit protein aggregation. Finally, IBT21 prevented cell death caused by chemically induced ER stress and a proteotoxin, an aggression-prone prion protein, indicating that IBT21, as a chemical chaperone, alleviates ER stress.

## Results

### Cell-based high-throughput screen for chemical chaperones

The UPR is mainly mediated by the following three transcription factors: XBP1 for the IRE1 pathway, ATF4 for the PERK pathway and ATF6 for the ATF6 pathway. Activation of each branch of the UPR can be reported by transcriptional reporters containing ER stress response element-2 (ERSE2) (*Kokame et al., 2001*), unfolded protein response element (UPRE) (*Yoshida et al., 2001*) and amino acid response element (AARE) (*Bruhat et al., 2000*), which are ATF6, XBP1 and ATF4 binding sequences, respectively. We established a UPR activation EGFP reporter (EUA-EGFP) cell line

containing ten tandem repeats of ERSE2 (10X <u>E</u>RSE2), ten tandem repeats of UPRE (10X <u>U</u>PRE) and five tandem repeats of AARE (5X <u>A</u>ARE) in the region upstream of a minimal promoter (*Figure 1A*). The EUA-EGFP reporter cells can simultaneously monitor the activation of the three branches of the UPR with high sensitivity and specificity using the specific ER stressor tunicamycin (Tm), an inhibitor of N-linked glycosylation (*Figure 1B* and *Figure 1—figure supplement 1*).

ER chemical chaperones are small molecules that prevent protein aggregation, thereby theoretically suppressing UPR activation under ER stress conditions. We hypothesized that chemical chaperones could be identified by monitoring UPR activation for reductions under ER stress conditions. The EUA-EGFP reporter cells allowed us to perform 384-well HTS via a sensitive (signal-to-background ratio = 5.09 ± 1.28) and robust (Z' factor = 0.78 ± 0.06) homogeneous assay using a microplate reader (*Figure 1C*). We screened the 217,765-compound chemical library from the Drug Discovery Initiative at the University of Tokyo. The small molecule inhibition of EUA-EGFP cells was normalized to the untreated control (assigned to be 100% inhibition), as was the cell viability, allowing comparisons between the screening plates. A first screening was carried out with all compounds at a concentration of 5 µM. We identified 1920 compounds for which the % EUA-EGFP inhibition was greater than the average + 5 SD of the Tm-treated control (hit cut-off = 37%) and the % viability was greater than the average + 3 SD of the Tm-treated control (hit cut-off = 78%) (*Figure 1D and E*). Confirmation screening was carried out with all compounds at a concentration of 0.5 µM in the second screening round. To decrease the number of compounds for follow-up, we chose the top 10 inhibition hits. A maximum common substructure search identified a common scaffold in 4 of 10 compounds: the 2-phenylimidazo[2,1-*b*]benzothiazole skeleton (shown in red) (*Figure 1D and F*). Although the other six compounds had the same high activity, they were all singletons with no common chemical structure. Therefore, we prioritized the analysis of the compounds with the 2-phenylimidazo[2,1-*b*]benzothiazole scaffold. The other six compounds remain to be analyzed in future studies.

## Structure-activity relationship study of imidazo[2,1-*b*]benzothiazole derivatives (IBTs)

To define the common scaffold of the hits, a structure-activity relationship (SAR) study was conducted for the 28 compounds considered imidazo[2,1-*b*]benzothiazole derivatives (IBTs). The 50% inhibitory concentration (IC$_{50}$) for UPR activation was calculated using a five-point dose-response curve for the 28 compounds. Hierarchical structure clustering analysis revealed four major clusters in the dendrogram (*Figure 2* and *Figure 2—figure supplement 1*). The active compounds belonged to two clusters, suggesting that both R1 and R2 in the IBTs are important for the observed activity. Removal of the phenyl group at the R1 position completely abolished UPR inhibition, indicating that the phenyl group is necessary for potency. Preferred substituents at the *p*-position of the R1 phenyl group include hydrogen groups (IBT6, IBT9, IBT15, IBT19 and IBT21), halogen groups (IBT2, IBT17 and IBT22), an ethyl group (IBT14) and a methoxy group (IBT3, IBT5, IBT8 and IBT12). In contrast, methyl groups (IBT4 and IBT10) and amino groups (IBT18) were not preferred. To investigate whether the activity decrease at R2 is caused by an amide bond or by bulkiness, we designed and synthesized a methyl-substituted amide-containing IBT (IBT29). IBT29 exhibited higher activity than IBT21 (*Figure 2—figure supplement 2*). Thus, for R2, a short substituent, particularly methylamide, endowed the scaffold with the desired activity. We observed that the preferred substituents at R2 are hydrogen groups (IBT1, IBT11, IBT12, IBT13, IBT17 and IBT20), methylsulphonyl groups (IBT21 and IBT22), a methoxycarbonyl group (IBT14 and IBT15) and a methylamide group (IBT29). Non-preferred substituent is carboxy group (IBT16).

For the next step, we needed to select one compound from the top five inhibitory IBTs (IBT14, IBT15, IBT17, IBT21 and IBT22) as a lead compound for further characterization of the IBT mechanism of action. One of the limitations of the current screening is that the target proteins of the hit compounds were unidentified. Since small molecule–target interaction profiles are useful for structure-based drug design to improve the lead compound, this compound should be capable of linker attachment to pull down a target protein. To obtain a bioactive IBT probe for pulldown, R1 could be an appropriate site for linker attachment because the SAR results predicted that linker attachment at the R2 position might lead to loss of activity. IBT21 is the only compound that has no substituent at the R1 phenyl group 2-position among the top five compounds. In terms of pharmacokinetics and bioavailability, a compound with a lower logP value is preferred, and IBT21 was the compound with

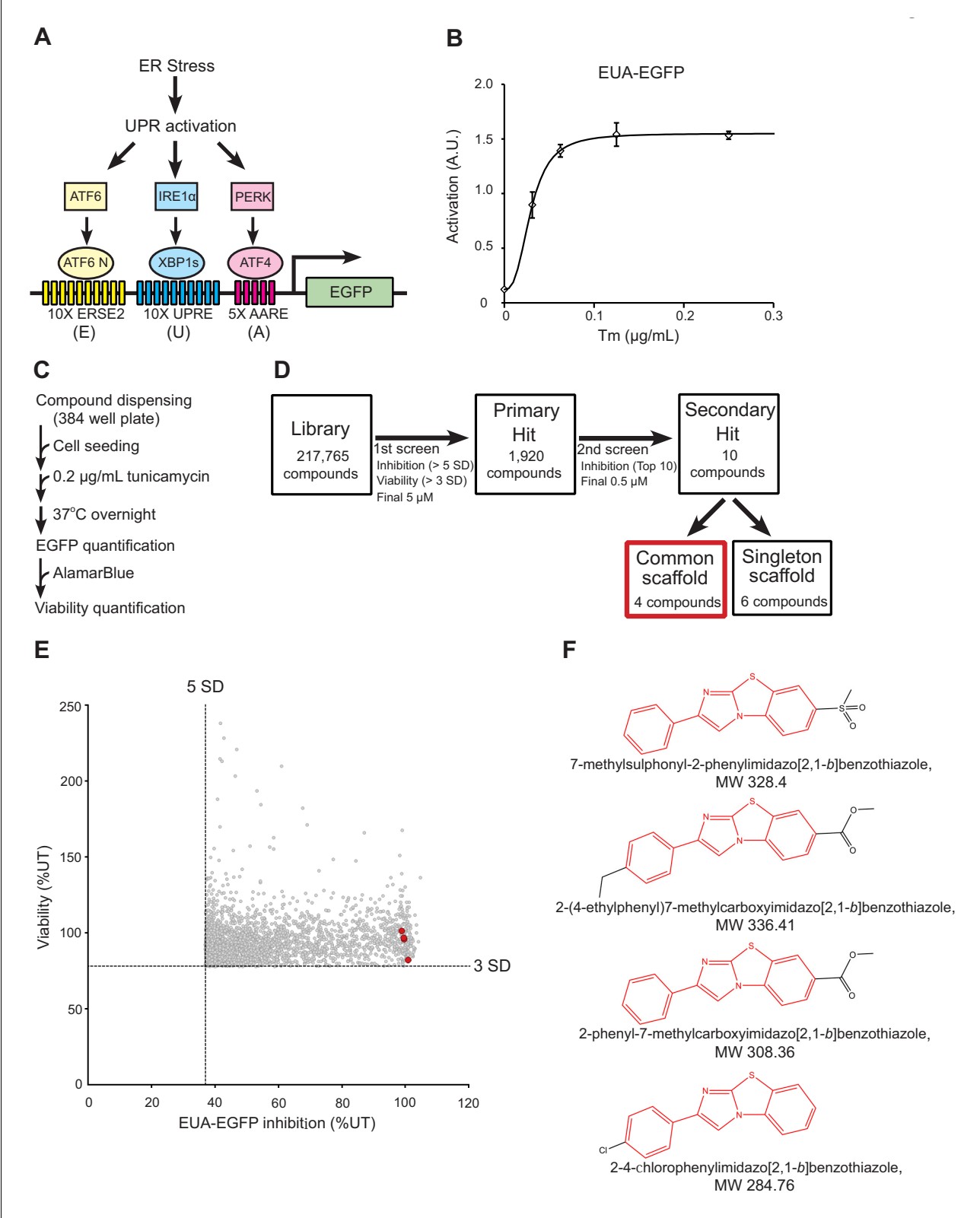

**Figure 1.** Cell-based high-throughput screening (HTS) for chemical chaperones. (**A**) Schematic of the UPR activation reporter (EUA-EGFP) in our HTS that can sense the three branches of the UPR using three stress response elements. ERSE2, ER stress response element-2; UPRE, unfolded protein response element; AARE, amino acid response element. (**B**) Activation of EUA-EGFP in HEK293A cells treated with the indicated concentration of tunicamycin (Tm) treatment overnight. Error bars show the mean ± SD (*n* = 4). (**C**) HTS protocol for chemical chaperone identification. (**D**) Flow diagram

*Figure 1 continued on next page*

*Figure 1 continued*

for the identification of chemical chaperones. The hit selection criteria and criteria used for prioritizing compounds are shown. (E) Results of the primary screen. Four compounds with a common scaffold among the top 10 hit compounds are colored red. (F) Chemical structure of the four compounds with a common scaffold among the top 10 hit compounds. The 2-phenylimidazo[2,1-*b*]benzothiazole skeletal formula, which was the common scaffold, is shown in red.

The online version of this article includes the following source data and figure supplement(s) for figure 1:

**Source data 1.** Dataset for *Figure 1*.
**Figure supplement 1.** Valildation of the UPR activation EGFP reporter (EUA-EGFP) cell line.
**Figure supplement 1—source data 1.** Dataset for *Figure 1—figure supplement 1*.

the lowest logP value. Given these considerations, we decided to further characterize IBT21, and synthesized IBT21 ourselves and confirmed its structure and activity.

## Comparison of IBT21 and the validated UPR modulator and chemical chaperone

Several compounds have been reported to mitigate ER stress, such as UPR modulators or chemical chaperones. To the best of our knowledge, IBT21 has a unique structure compared with previously reported UPR modulators or chemical chaperones (*Figure 3A*). According to a database search, IBT21 was registered in PubChem (CID: 735314, https://pubchem.ncbi.nlm.nih.gov/) and SciFinder (CAS Registry Number 692744-81-3, https://scifinder.cas.org/). Moreover, there were 3042 types of 2-phenyl IBT scaffold compounds in PubChem and 9878 types in SciFinder as of February 24, 2019, but no chemical chaperone-like activity has been reported. To assess the effects of IBT21, we next aimed to compare the activity of IBT21 to that of a validated UPR modulator and a chemical chaperone, azoramide (AZO) and 4PBA, respectively. To evaluate the activity with more information including the cell morphology, we first established a UPR inhibition assay using the EUA-EGFP reporter and high content image analysis. Cell morphology and EUA-EGFP reporter activation can be evaluated simultaneously with high sensitivity (signal-to-background ratio = 15.3) and robustness (Z' factor = 0.75) (*Figure 3B,C and D*). At 10 µM IBT21, Tm-induced UPR activation was inhibited 91.8% by IBT21 in 293A cells (*Figure 3B–D*), as well as in another human cell line, Hap1 cells (*Figure 3—figure supplement 1*). We utilized our high content image analysis to investigate the potential UPR modulation by small molecules. IBT21 demonstrated an $IC_{50}$ value of 0.95 µM, consistent with the $IC_{50}$ (0.61 µM) obtained from the HTS assay (*Figure 3E*). However, compared with IBT21, AZO and 4PBA showed much higher $IC_{50}$ values of 86 µM and greater than 5,000 µM, respectively (*Figure 3E*). We concluded that our phenotypic screening approach identified a new and more potent small molecule, IBT21.

## Inhibitory effects of IBT21 on activation of the three UPR branches under ER stress conditions

To rule out the possibility that IBT21 exerts its UPR inhibitory effect by suppressing one major branch of the UPR, we explored its effects on the three UPR branches. To directly monitor each branch of the UPR, we established EGFP reporter cell lines containing twenty-five tandem repeats of ERSE2, UPRE or AARE in the region upstream of a minimal promoter to detect the activation of the ATF6, IRE1 or PERK branches of the UPR, respectively. As expected, IBT21 suppressed the Tm-induced activation of all three reporters in a dose-dependent manner (*Figure 4A–C*). Indeed, the $IC_{50}$ values for ERSE2 (0.24 µM), UPRE (0.33 µM) and AARE (0.46 µM) were similar to that of EUA-EGFP (0.61 µM), producing a similar dose-response curve. To further rule out off-target effects through which IBT21 might induce a pseudo-response to the reporter, we assessed the effects of IBT21 by examining the expression levels of UPR downstream target genes by immunoblot analyses. As expected, the levels of ATF4, GADD34 and CHOP, markers of PERK branch signalling, were increased by Tm, and IBT21 treatment abolished these increases in a dose-dependent manner (*Figure 4D,E* and *Figure 4—figure supplement 1*). Additionally, IBT21 decreased the levels of DNAJB9, a marker of IRE1 branch signalling, and SEL1L and HERPUD1, markers of ATF6 branch signalling, which were induced by Tm. By contrast, the UPR modulator AZO did not decrease any

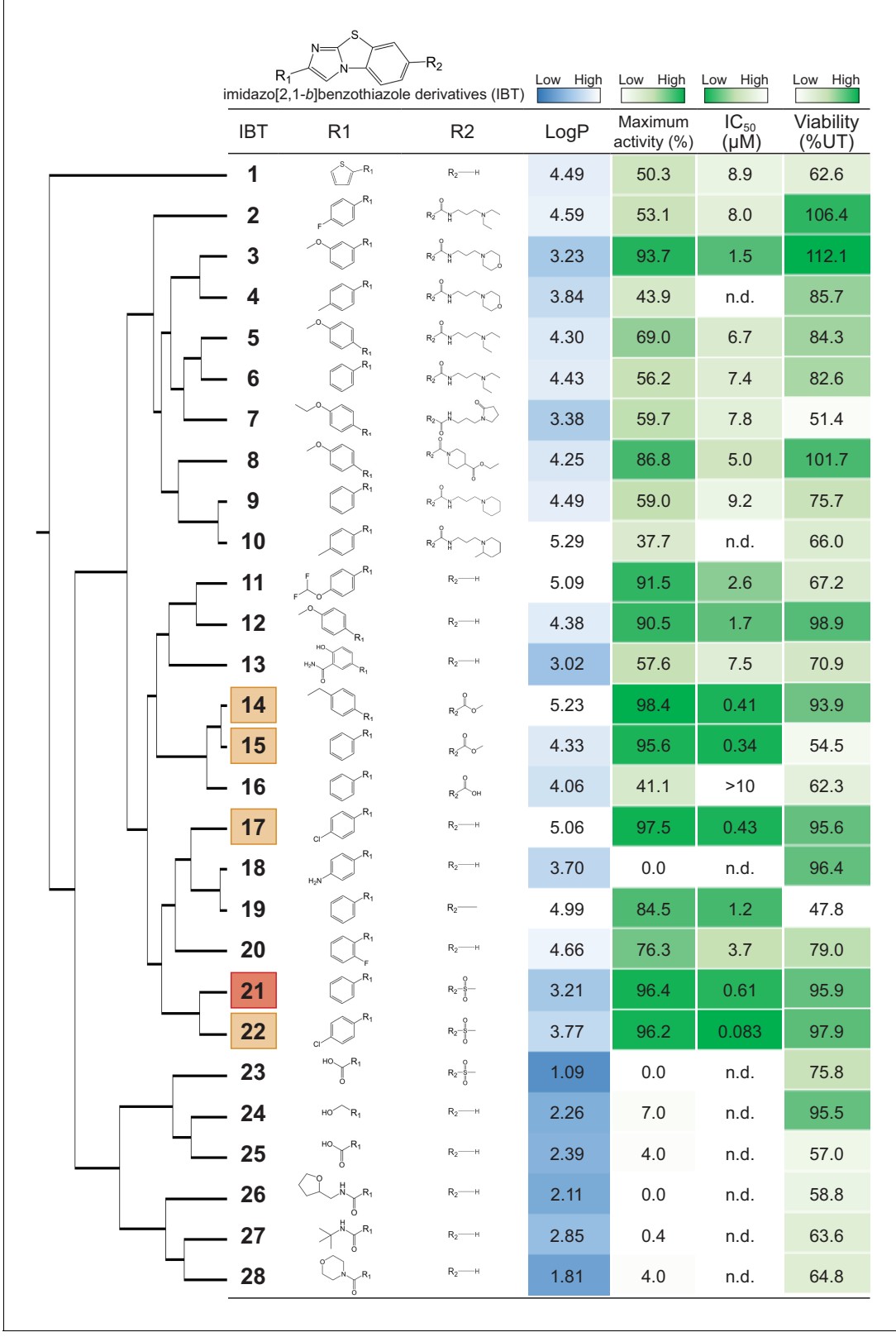

**Figure 2.** Structure-activity relationship study of imidazo[2,1-*b*]benzothiazole derivatives (IBTs). The dendrogram for the hierarchical structural clustering of 28 IBTs is shown with the LogP, maximum activity (%), IC$_{50}$ (µM) of EUA-EGFP and viability (%) of the cells treated with 0.2 µg/mL Tm overnight relative to that of the untreated cells. The top compound (IBT21) is highlighted in red and the 2nd to 5th compounds are highlighted in orange. n.d.: not-determined.

*Figure 2 continued on next page*

*Figure 2 continued*

The online version of this article includes the following source data and figure supplement(s) for figure 2:

**Figure supplement 1.** Inhibition of the EUA-EGFP reporter in HEK293A cells treated with 0.2 µg/mL Tm in the presence of IBTs at 5- or 6-point dose.
**Figure supplement 1—source data 1.** Dataset for *Figure 2—figure supplement 1*.
**Figure supplement 2.** IBT29 exhibited higher activity than IBT21.
**Figure supplement 2—source data 1.** Dataset for *Figure 2—figure supplement 2*.

markers in the UPR by Tm, consistent with the previous report (*Fu et al., 2015*). These results indicated that IBT21 works as a chemical chaperone rather than a UPR modulator.

However, one might say that IBT21 may influence many of the observed readouts by reducing transcription/translation, rather than by functioning as a chemical chaperone. To demonstrate the chemical chaperone activity of IBT21, we conducted several experiments. First, we overexpressed active UPR transcription factor and examined the effects of IBT21 on transcriptional induction of UPR target genes (*Figure 4—figure supplement 2*). IBT21 did not change the expression of CHOP or 4E-BP1, downstream targets of ATF4, in the ATF4-overexpressing cells. Additionally, IBT21 did not change the expression of BiP in the cells overexpressing an active form of ATF6 [ATF6 (N)], These data show that IBT21 does not affect the ability to induce UPR target genes. Next, we assessed the effects of IBT21 on protein translation by measuring protein synthesis based on the incorporation of puromycin into newly synthesized proteins through its detection with anti-puromycin antibodies (*Figure 4—figure supplement 3*). We evaluated that IBT21 does not affect protein translation. Furthermore, we tested whether IBT21 can also reduce ER stress caused by dithiothreitol (DTT), which does not require protein translation to induce ER stress (*Figure 4—figure supplement 4*). We showed that IBT21 modestly reduced DTT-induced ER stress by monitoring the activation of EUA-EGFP reporter cells. In contrast, IBT21 could not suppress the induction of heat-shock response by heat shock (*Figure 4—figure supplement 5*). These data support a chemical chaperone function of IBT21 during ER stress but not all proteostasis stress.

## In vivo effects of IBT21 on protein aggregation during ER stress

To directly address whether IBT21 exerted its effects though chemical chaperone activity, we investigated the effects of IBT21 on reducing protein aggregation during ER stress. To monitor protein aggregation in vivo, we first took advantage of the ProteoStat dye, a molecular rotor that binds selectively to aggregated proteins (*Shen et al., 2011*). Proteasomal inhibition is known to lead to the accumulation of ubiquitin-conjugated proteins organized in perinuclear structures termed 'aggresomes'. As expected, aggregated protein cargo was visualized with a punctate red fluorescence pattern in cells treated with the proteasome inhibitor MG132. When we treated cells with Tm, diffuse cytoplasmic staining was observed that differed between the control cells and MG132-treated cells with discrete punctuate staining, indicating the accumulation of aggregated proteins in the ER (*Figure 5A*). Quantification of the ProteoStat dye staining within cells revealed increased fluorescence intensity in the Tm-treated cells (2.29-fold of untreated (UT)) and MG132-treated cells (2.33-fold of UT), and lower fluorescence intensity in the cells co-treated with Tm and IBT21 (0.87-fold of UT), showing that IBT21 prevented protein aggregation induced by Tm (*Figure 5B*). To validate the direct effect of IBT21 on protein aggregation inhibition, an in vitro protein aggregation assay was performed using human insulin and the ProteoStat dye. Human insulin was challenged to be denatured by overnight incubation of DTT with or with IBT, 4PBA or AZO. As expected, IBT21 and 4PBA inhibited protein aggregation, indicating a direct function as chemical chaperones, though the UPR modulator AZO did not (*Figure 5C*). Notably, IBT21 inhibited protein aggregation at a concentration of 10 µM, which is 100-fold lower than that of 4PBA.

For further assessment of IBT21, we performed a biochemical assay to detect misfolded or aggregated proteins. Misfolded or aggregated proteins are prone to forming stable, detergent-insoluble, high molecular weight (HMW) oligomers together with molecular chaperones (*Johnston et al., 1998*). We previously developed and modified a method to estimate the amount of misfolded or aggregated protein in the ER based on the levels of the molecular chaperone BiP (also known as HSPA5) in SDS-resistant HMW complexes (**AGGREGATED**) (*Marciniak et al., 2004*; *Hisanaga et al., 2018*) (*Figure 5D*). Compared with the levels of BiP in the INPUT fraction, the levels of BiP in the

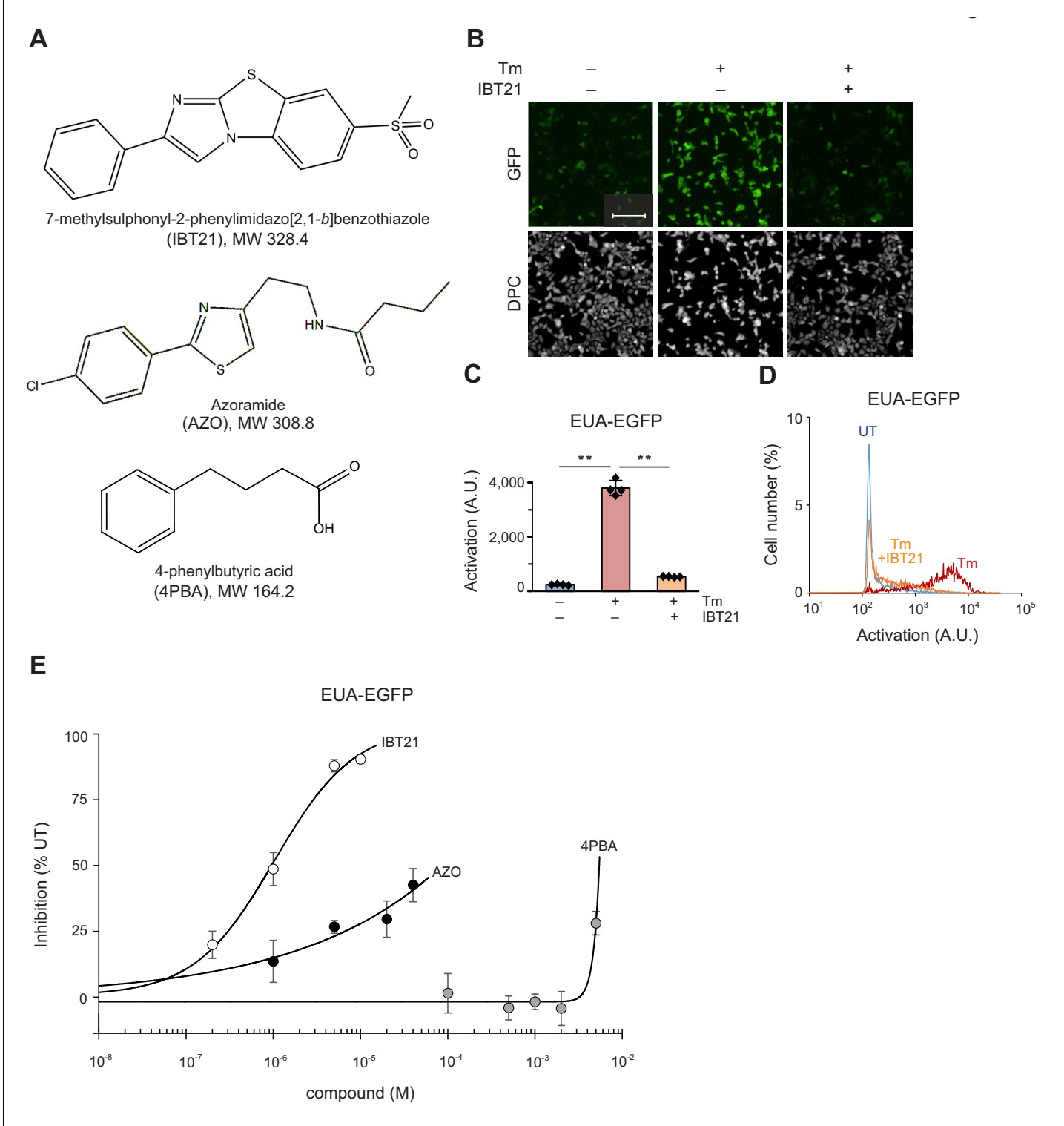

**Figure 3.** Comparison of IBT21 and the validated UPR modulator and chemical chaperone. (**A**) Chemical structure of IBT21, azoramide (AZO) and 4PBA. (**B**) Representative images of HEK293A-EUA-EGFP cells treated with 0.2 μg/mL Tm in the presence or absence of 10 μM IBT21. DPC: digital phase contrast. The scale bar represents 200 μm. (**C**) GFP fluorescence intensities of the EUA-EGFP reporter with 0.2 μg/mL Tm in the presence or absence of 10 μM IBT21. Error bars show the mean ± SD ($n = 4$). Data were analyzed using one-way ANOVA with Tukey's post hoc test, **$p<0.01$. (**D**) Distribution of GFP fluorescence from the EUA-EGFP reporter with 0.2 μg/mL Tm in the presence or absence of 10 μM IBT21. (**E**) Inhibition of the EUA-EGFP reporter in HEK293A cells treated with 0.2 μg/mL Tm in the presence of IBT21, AZO or 4PBA. Error bars show the mean ± SD ($n = 4$).

*Figure 3 continued on next page*

*Figure 3 continued*

The online version of this article includes the following source data and figure supplement(s) for figure 3:

**Source data 1.** Dataset for *Figure 3*.

**Figure supplement 1.** Tm-induced UPR activation was inhibited by IBT21 also in another human cell line Hap1.

**Figure supplement 1—source data 1.** Dataset for *Figure 3—figure supplement 1*.

AGGREGATED fraction corresponding to misfolded or aggregated proteins in the ER were increased by Tm treatment (3.1-fold of UT), and this increase was significantly suppressed by IBT21 treatment (1.7-fold of UT) (*Figure 5E and F*). IBT21 treatment did not change the levels of BiP under ER stress conditions (*Figure 5E and F*) or under basal conditions (*Figure 4—figure supplement 2C, D*), indicating that IBT21 did not lead to reduced expression or enhanced degradation of BiP. Having established that IBT21 reduced the levels of the SDS-resistant HMW BiP-containing complex induced by Tm, we next determined the profile of proteins in the SDS-resistant HMW BiP-containing complex to directly monitor proteostasis changes. The AGGREGATED fraction samples containing the SDS-resistant HMW BiP-containing complex were subjected to mass spectrometry analysis, and 1233 proteins were detected by this LC-MS/MS analysis. To avoid false positive protein identification, we filtered 598 proteins identified with $\geq$2 unique peptides per protein and $\geq$5 peptide-spectrum matches. Then, to determine the proteins modulated by IBT21, we analyzed the differentially expressed proteins among the 598 proteins with the following criteria: 1) 1.5-fold enrichment in the Tm-treated samples compared to the UT samples and 2) 1.5-fold enrichment in the Tm-treated samples compared to the Tm- and IBT21-treated samples. We found that 211 of 598 proteins (35.3%) were affected by Tm, and 125 of those 211 proteins (59.2%) were affected by IBT21 (IBT-SENSITIVE proteins), suggesting that IBT21 prevents Tm-induced protein misfolding (*Figure 5G*, *Figure 5—figure supplement 1A*). Furthermore, their subcellular localization and cellular functions were classified by Gene Ontology (GO) annotation. Extracellular compartments were highly enriched in the category of cellular components (*Figure 5H Figure 5—figure supplement 1B*). These enrichments support that IBT21 acts on the folding of proteins synthesized in the ER because secreted and membrane-bound proteins are synthesized in the ER. For example, the proteins we identified included Annexin A1 (ANXA1) and Transgelin-2 (TAGLN2), and these proteins represent the protein group affected by Tm and IBT21, as ANXA1 (*Goulet et al., 1992*) and TAGLN2 (*Liu et al., 2018*) were reported or predicted to have *N*-linked glycosylation sites for secretion (*Figure 5G* and *Figure 5—figure supplement 1A*). In addition to ER client proteins, proteins required for proteostasis could also be affected by Tm and IBT21. As expected, protein folding was highly enriched in the category of biological function, as the representative proteins from the AGGREGATED fraction included the molecular chaperone HSPA5 and the ER folding enzyme PDIA4 (*Figure 5G,I*, *Figure 5—figure supplement 1A,C*). Thus far, our observations support that IBT21 has chemical chaperone activity against ER stress.

## In vivo target protein identification of IBT21 during ER stress

To examine the direct binding of IBT21 to unfolded or misfolded proteins in vivo, we took advantage of diazido photoaffinity labelling probes, which allow a chemical biology approach for profiling IBT21-binding proteins (*Hosoya et al., 2004*; *Hosoya et al., 2005*). We designed a diazido-functionalized IBT21 analogue, diazido-IBT21, as a candidate photoaffinity labelling probe (*Figure 6A*). The activity of diazido-IBT21 was comparable to that of IBT21, and we used the more potent compound IBT22 as a competitor (*Figure 6B*). With a promising diazido-IBT21 probe in hand, we performed in vivo target protein identification experiments (*Figure 6C*). In principle, the target proteins were specifically captured by covalent bonding via photoreaction of the probe aromatic azido group. The remaining probe aliphatic azido group was subsequently subjected to biotin labelling via Cu-catalysed azide alkyne cycloaddition (CuAAC) for pulldown of the target proteins. In total, 1441 proteins were detected as IBT-BOUNDED proteins and were classified with the following criteria: 1) 1.5-fold change in the probe-treated samples compared to the probe- and competitor-treated samples (competed) and 2) 1.5-fold change in the probe-treated samples compared to the UT samples (selective) (*Figure 6D*). The IBT-BOUNDED proteins were categorized into the following four groups: Competed and Selective (1205 proteins), Competed (11 proteins), Selective (185 proteins)

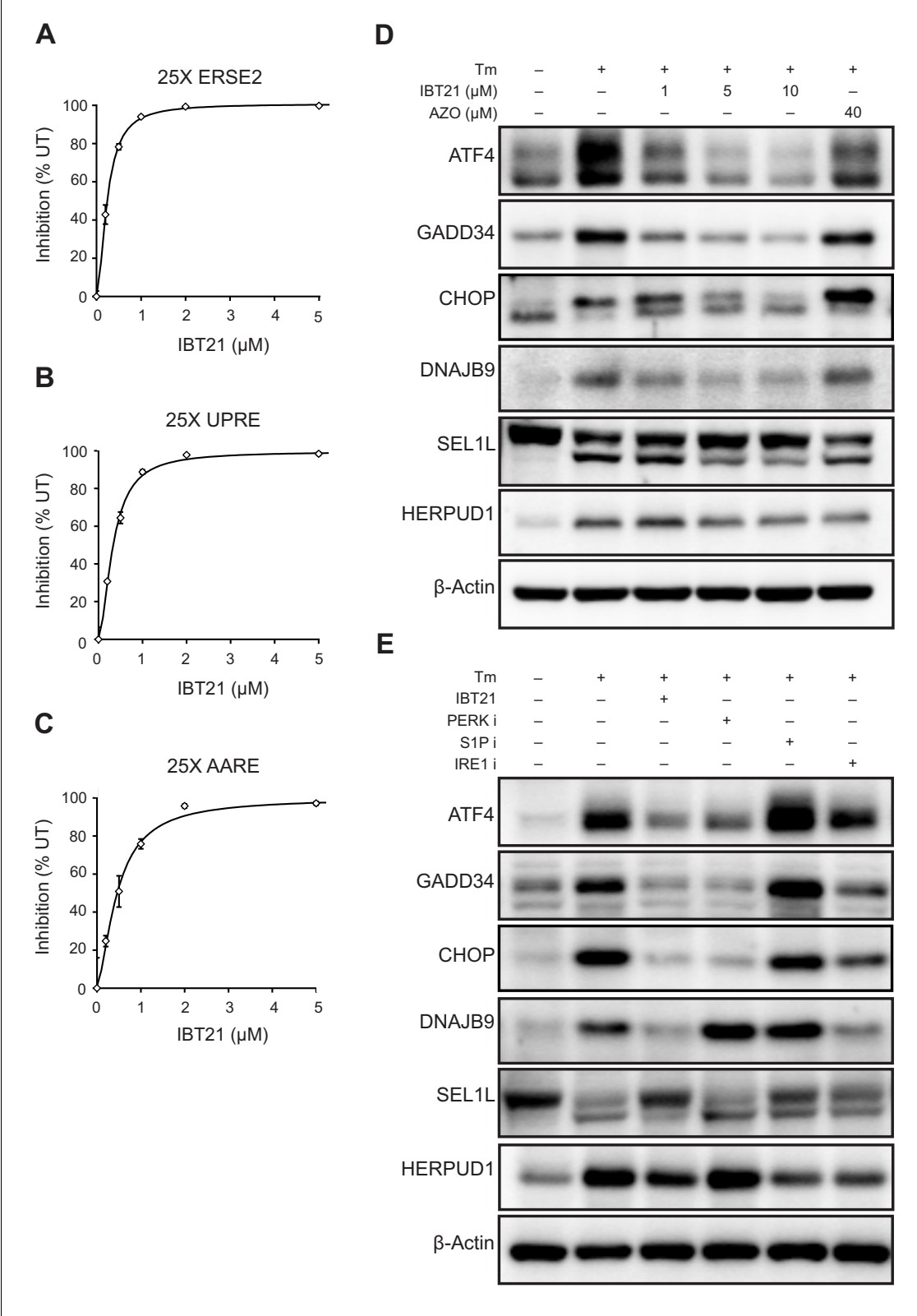

**Figure 4.** Inhibitory effects of IBT21 on the activation of the three UPR branches under ER stress conditions. (A–C) Inhibition of the 25X ERSE2-EGFP (A), 25X UPRE-EGFP (B) or 25X AARE-EGFP (C) reporters in HEK293A cells treated with 0.2 μg/mL Tm in the presence or absence of IBT21. Error bars show the mean ± SD ($n$ = 4). (D) Representative immunoblots of ATF4, GADD34, CHOP, DNAJB9, SEL1L and HERPUD1 in HEK293A cells treated with 0.2 μg/mL Tm overnight in the presence or absence of IBT21 or AZO. β-actin was used as a loading control. (E) Representative immunoblots of ATF4,

*Figure 4 continued on next page*

*Figure 4 continued*

GADD34, CHOP, DNAJB9, SEL1L and HERPUD1 in HEK293A cells treated with 0.2 µg/mL Tm overnight in the presence or absence of 10 µM IBT21, 1 µM PERK inhibitor I (GSK2606414), 200 µM IRE1 inhibitor I (STF083010) or 2 µM Site-1-protease inhibitor (PF429242). β-actin was used as a loading control.

The online version of this article includes the following source data and figure supplement(s) for figure 4:

**Source data 1.** Dataset for *Figure 4*.
**Figure supplement 1.** IBT21 treatment abolished Tm-induced induction of UPR downstream target genes.
**Figure supplement 1—source data 1.** Dataset for *Figure 4—figure supplement 1B*.
**Figure supplement 2.** IBT21 does not affect the ability to induce UPR target genes.
**Figure supplement 2—source data 1.** Dataset for *Figure 4—figure supplement 1*.
**Figure supplement 3.** IBT21 does not affect protein translation.
**Figure supplement 3—source data 1.** Dataset for *Figure 4—figure supplement 3*.
**Figure supplement 4.** IBT21 can also reduce ER stress caused by dithiothreitol (DTT).
**Figure supplement 4—source data 1.** Dataset for *Figure 4—figure supplement 4*.
**Figure supplement 5.** IBT21 could not suppress the induction of heat-shock response by heat shock.
**Figure supplement 5—source data 1.** Dataset for *Figure 4—figure supplement 5*.

and Non-Specific (40 proteins) (*Figure 6E*). The **Competed and Selective prot**eins were the majority (84%) of the IBT-**BOUNDED** proteins. To characterize the IBT21-targeted proteins to corresponding chemical chaperone activity, we analyzed the **Overlapped** proteins observed in both the IBT-**SENSITIVE** proteins and the **Competed and Selective** proteins. As expected, the Overlapped proteins included ANXA1 and TAGLN2, which were also identified as IBT21-rescued aggregation-prone proteins (IBT-**SENSITIVE** proteins), as shown in *Figure 5* (*Figure 6G*). The GO enrichment analysis revealed that groups of extracellular components were highly enriched in the **Overlapped** proteins (*Figure 6—figure supplement 1A*). GO enrichment results of the IBT-**SENSITIVE** proteins were different from that of the IBT-bonded **Competed and Selective** proteins (*Figure 6—figure supplement 1B–C*) because IBT21 is stabilizing many more proteins that cannot be detected in the experiment that determined IBT-sensitive compounds. Nevertheless, the **Overlapped** proteins contained approximately 60% of the IBT-**SENSITIVE** proteins in each enriched GO group, indicating that the **Competed and Selective** proteins largely cover the IBT21-targeted proteins corresponding to chemical chaperone activity (*Figure 6H*). Taken together, these results support that IBT21 directly binds unfolded or misfolded proteins to prevent protein aggregation in vivo.

## Cytoprotective effect of IBT21 during ER stress

We finally tested whether IBT21 could ameliorate ER stress. We first evaluated the effects of IBT21 on cell proliferation under ER stress conditions, as measured by cell confluency using the high content imaging system. Tm treatment almost completely inhibited cell proliferation, as was observed in cells treated with the cytotoxic alkaloid camptothecin (cell death control). Treatment with 1 µM IBT21 restored the cell growth inhibited by Tm up to 69.8% of the growth of the UT cells (*Figure 7A and B*). Surprisingly, cell growth was significantly stimulated to 112.7 or 115.8% of that of UT cells by the addition of 5 or 10 µM IBT21, respectively, even in the presence of Tm. In contrast, 40 µM AZO treatment led to a significant but modest growth recovery up to 58.1% of that of the UT cells. We next evaluated the effects of IBT21 on cell viability or cytotoxicity as measured by intracellular ATP levels and lactate dehydrogenase release. The cytoprotective effects of IBT21 were also evident in biochemical measurements. Consistent with the results of the proliferation assays, IBT21 treatment attenuated viability loss and cell death in a dose-dependent manner (*Figure 7C and D*).

To corroborate the cytoprotective effects of IBT21, we next evaluated the effects of IBT21 against an ER proteotoxin. Prion disease is a neurodegenerative disorder characterized by the accumulation of an abnormally folded and protease-resistant form of prion protein (PrP). As a secretory protein, PrP is synthesized in the ER, and approximately 10% of PrP is naturally misfolded (*Yedidia et al., 2001*). Accumulating evidence suggests that perturbations in ER homeostasis may contribute to neurodegeneration in prion disease (*Shi and Dong, 2011*; *Torres et al., 2015*). Therefore, we chose mutant PrP, which was previously reported to cause ER stress, as an ER proteotoxin (*Satpute-Krishnan et al., 2014*). To monitor the expression levels of PrP, EGFP was fused to wild-type PrP (WT PrP) or mutant PrP (mut PrP). Overexpression of mut PrP inhibited cell proliferation, and 5 µM

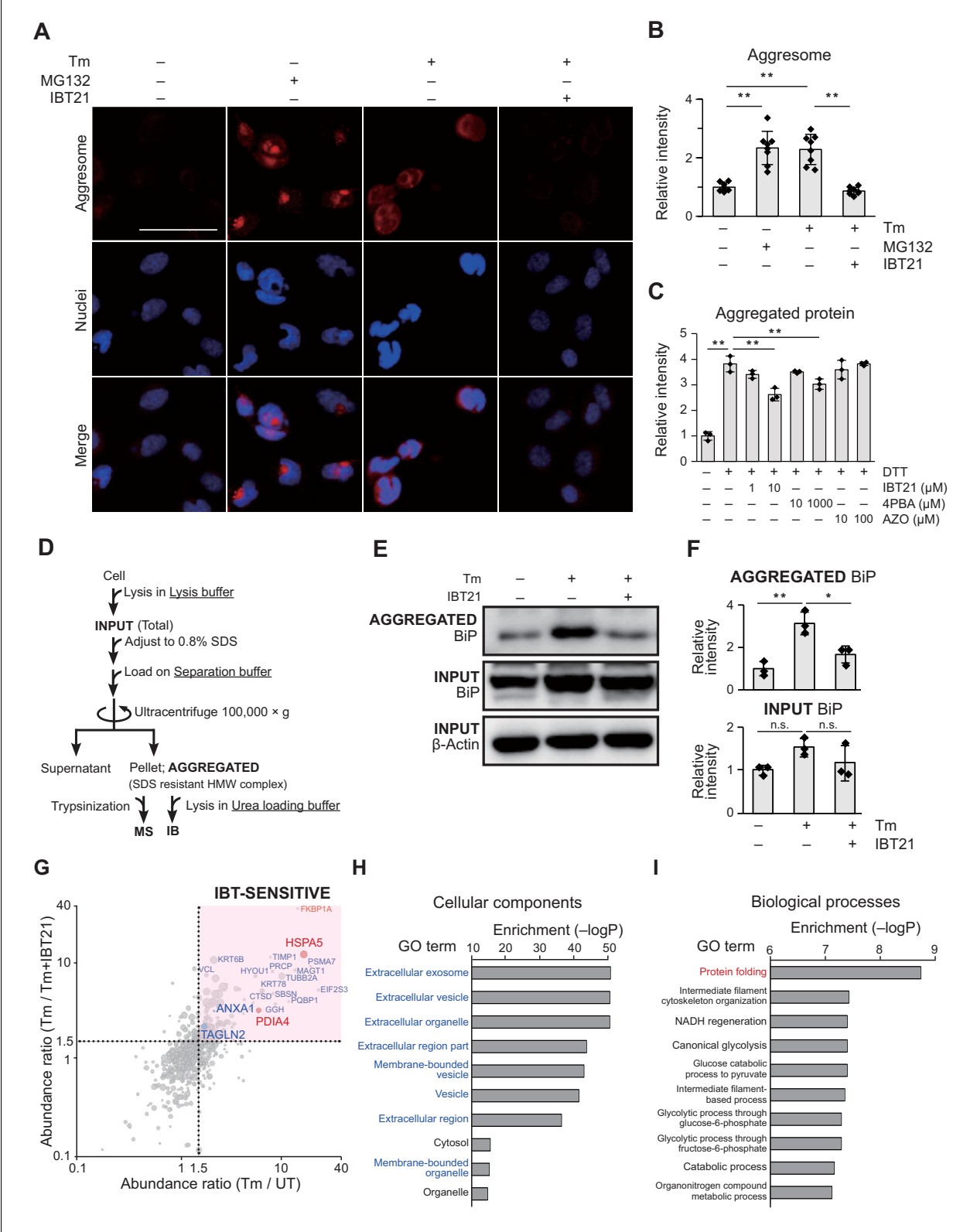

**Figure 5.** In vivo effects of IBT21 on protein aggregation during ER stress. (**A**) Representative immunofluorescence images of protein aggregates stained with ProteoStat dye in HEK293A-EUA-EGFP cells treated with 0.2 µg/mL Tm in the presence or absence of 10 µM IBT21. MG132 was used as an aggregation control. The scale bar represents 50 µm. (**B**) Relative fluorescence intensities of aggresome staining in A. Error bars show the mean ± SD (*n* = 8). One-way ANOVA with Tukey's post hoc test, **p<0.01. (**C**) Relative fluorescence intensities of insulin aggregates induced by DTT treatment and

*Figure 5 continued on next page*

*Figure 5 continued*

stained with ProteoStat dye with or without the indicated concentration of IBT21, 4PBA or AZO. Error bars show the mean ± SD (*n* = 3). One-way ANOVA with Tukey's post hoc test, **p<0.01. (**D**) Protocol for high molecular weight (HMW) detergent-resistant complex separation. HEK293A cells treated with 0.2 µg/mL Tm in the presence or absence of 10 µM IBT21 were lysed in lysis buffer, adjusted to 0.8% SDS, loaded on separation buffer and then ultracentrifuged at 100,000 × g for 55 min. The pellet fraction (SDS-resistant HMW complex) was considered aggregated proteins (**AGGREGATED**). (**E**) Representative immunoblots of BiP in the **AGGREGATED** SDS-resistant HMW complexes and in the **INPUT**. (**F**) Densitometry quantification of the immunoblot analysis in E. The protein levels of BiP in the **AGGREGATED** and **INPUT** were normalized to β-actin levels in the **INPUT**. Error bars show the mean ± SD (*n* = 3). One-way ANOVA with Tukey's post hoc test, *p<0.05, **p<0.01. n.s: not significant. (**G**) Bubble plot representation of HMW detergent-resistant aggregated proteins. The X-axis denotes the Tm to UT abundance ratio in the absence of IBT21. The Y-axis denotes the Tm to Tm+IBT21 abundance ratio. Each threshold is 1.5-fold-change. Representative hit proteins, such as HSPA5, PDIA4, ANXA1 and TAGLN2, are indicated. Peptide-spectrum matches (PSMs) are correlated with the size of the circle. (**H–I**) Functional classification of proteins protected from aggregation by IBT21. Aggregated proteins that were suppressed by IBT21 (Tm/UT > 1.5 and Tm/Tm+IBT21 >1.5) were categorized on the basis of cellular components (**H**) and biological processes (**I**) through Gene Ontology (GO) annotation.

The online version of this article includes the following source data and figure supplement(s) for figure 5:

**Source data 1.** Dataset for *Figure 5B, C and F*.
**Source data 2.** Dataset for *Figure 5G*.
**Source data 3.** Dataset for *Figure 5H*.
**Source data 4.** Dataset for *Figure 5I*.
**Figure supplement 1.** IBT21 prevented Tm-induced protein misfolding.
**Figure supplement 1—source data 1.** Dataset for *Figure 5—figure supplement 1A*.
**Figure supplement 1—source data 2.** Dataset for *Figure 5—figure supplement 1B*.
**Figure supplement 1—source data 3.** Dataset for *Figure 5—figure supplement 1C*.

IBT21 treatment restored the cell growth inhibition caused by mutant PrP overexpression (*Figure 7E and F*). Consistent with the results from the proliferation assays using the high content imaging system, the recovery of viability loss by IBT21 treatment in the mutant PrP-overexpressing cells was also confirmed by a biochemical assay with CellTiter-Glo. Collectively, these data provide in vivo evidence that IBT21 protects cells against ER stress conditions.

## Discussion

In the current study, we aimed to identify a potent chemical chaperone through the HTS of a small molecule chemical library. We discovered that IBTs function as chemical chaperones and directly bind unfolded or misfolded proteins to inhibit protein aggregation using the compound IBT21. Consistent with the protective nature of chemical chaperones, IBT21 protected cells from death caused by ER stress. The observed effects cannot be explained by a simple mechanism because these compounds have a planar conjugated ring system similar to many DNA intercalators; however, IBTs are promising scaffolds for chemical chaperones. Furthermore, based on our SAR studies, we might predict more potent IBT compounds. The combination of the methoxy group at the *m*-position of IBT3 or the chloro group at the *p*-position of IBT17 for R1 and the methoxycarbonyl group of IBT15 or methylamide group of IBT29 for R2 seems to be ideal. In addition to assessing the absorption, distribution, metabolism, and excretion properties of IBTs, we will pursue lead compound optimization in future studies.

Chemical chaperones can be divided into the following two groups: osmoprotectants and pharmacological chaperones. Osmoprotectants are generally considered to stabilize protein structure by altering solvent properties. The major osmoprotectants can be categorized in the following three chemical classes: polyols, such as glycerol and sucrose; free amino acid derivatives, such as taurine and β-alanine; and methyl-amines, such as DMSO and trimethylamine N-oxide (TMAO) (*Cortez and Sim, 2014*). Among these osmoprotectants, TMAO is the most efficient and has been the most studied over the past two decades. For example, TMAO has been shown to prevent the conversion of the normal cellular form of prion protein (PrP[C]) into its infectious form (PrP[SC]) in scrapies-infected mouse neuroblastoma ScN2a cells (*Tatzelt et al., 1996*). However, osmoprotectants are effective at high concentrations which may lead to toxicity. Thus, despite the high efficiency of the anti-aggregation properties of osmoprotectants, very few osmoprotectants have been applied in clinical trials. Another class of chemical chaperones is pharmacological chaperones, which are able to bind to and stabilize unfolded or misfolded proteins. Structurally, IBT21 was not particularly similar to

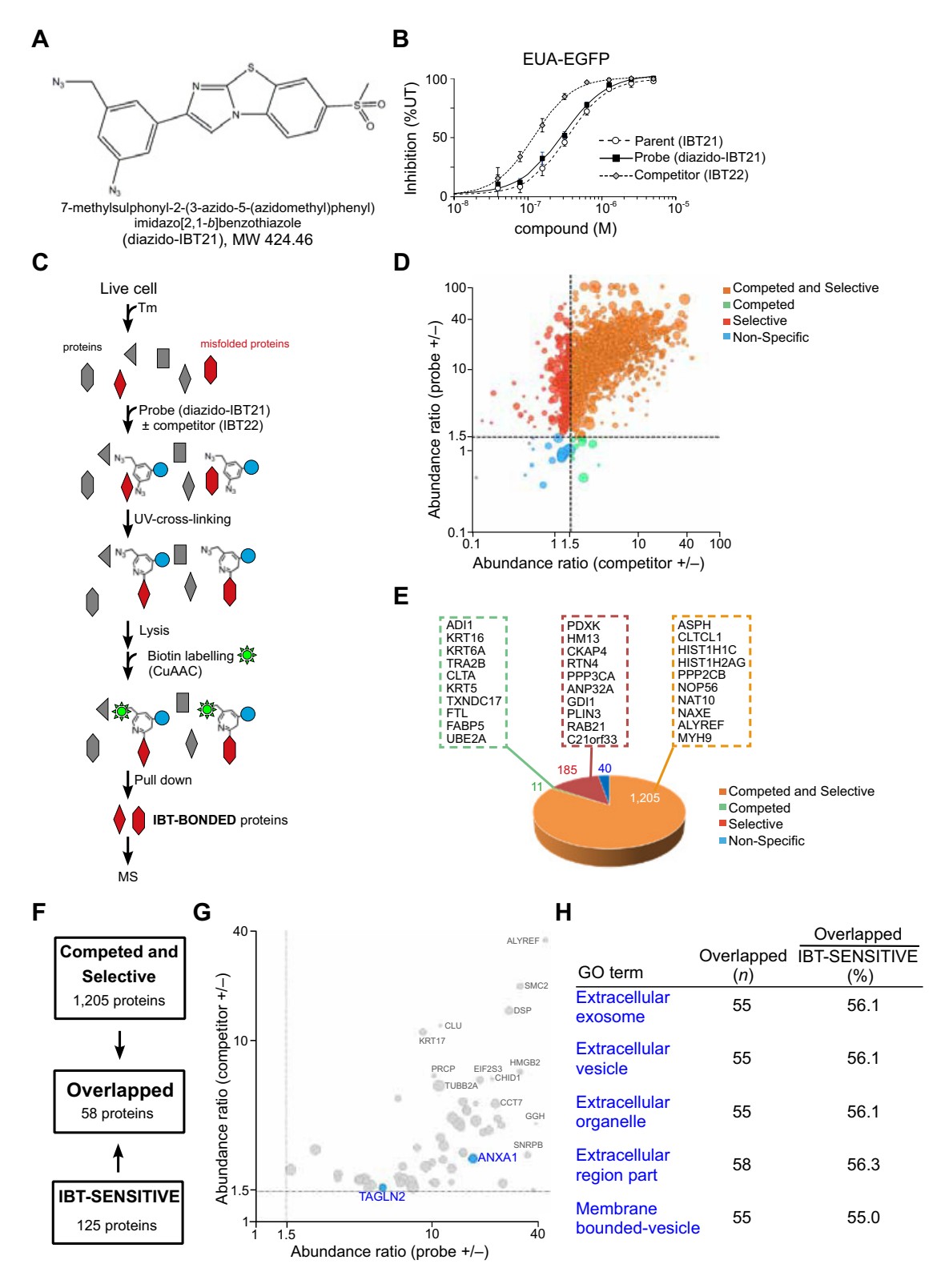

**Figure 6.** In vivo target protein identification of IBT21 during ER stress. (**A**) Chemical structure of the diazido-IBT21 probe. (**B**) Inhibition of the EUA-EGFP reporter in HEK293A cells treated with 0.2 µg/mL Tm in the presence of IBT21, diazido-IBT21 or IBT22. Error bars show the mean ± SD (*n* = 4). (**C**) Protocol for IBT-**BONDED** protein identification. HEK293A cells treated with 0.2 µg/mL Tm and 1 µM diazido-IBT21 probe were crosslinked to interacting proteins and labelled with biotin. (**D**) Bubble plot representation of the **IBT-BONDED** proteins. The X-axis denotes the abundance ratio of

*Figure 6 continued on next page*

*Figure 6 continued*

IBT competitor to no competitor. The Y-axis denotes the abundance ratio of diazido-IBT21 probe to no probe. Each threshold represents a 1.5-fold-change. Peptide-spectrum matches (PSMs) are correlated with the size of the circle. (E) Pie chart representation for the categorization of the **IBT-BONDED** proteins identified in experiments comparing cells treated with probe/no-probe and probe/probe + competitor. Each threshold represents a 1.5-fold-change. (F) Flow diagram for the comparison between the **IBT-SENSITIVE** and **Competed and Selective** proteins. (G) Bubble plot representation of the **Overlapped** proteins. The X-axis denotes the abundance ratio of diazido-IBT21 probe to no probe. The Y-axis denotes the abundance ratio of IBT competitor to no competitor. Each threshold represents a 1.5-fold-change. Representative hit proteins, such as ANXA1 and TAGLN2, are indicated in blue. Peptide-spectrum matches (PSMs) are correlated with the size of the circle. (D) Functional classification of the **Overlapped** proteins.

The online version of this article includes the following source data and figure supplement(s) for figure 6:

**Source data 1.** Dataset for *Figure 6*.
**Source data 2.** Dataset for *Figure 6D and E*.
**Source data 3.** Dataset for *Figure 6G*.
**Source data 4.** Dataset for *Figure 6H*.
**Figure supplement 1.** The Competed and Selective proteins largely cover the IBT21-targeted proteins corresponding to chemical chaperone activity.
**Figure supplement 1—source data 1.** Dataset for *Figure 6—figure supplement 1A*.
**Figure supplement 1—source data 2.** Dataset for *Figure 6—figure supplement 1B*.
**Figure supplement 1—source data 3.** Dataset for *Figure 6—figure supplement 1C*.

osmoprotectants, and IBT21 was observed to preferably bind to unfolded or misfolded proteins in combined analyses such as the ER stress-induced aggregated protein profile and the IBT21-targeted protein profile. Therefore, we considered that IBT21 acted as a pharmacological chaperone. In unfolded or misfolded proteins, the hydrophobic region tends to be exposed, and protein aggregation occurs when the exposed hydrophobic portions of a protein interact with the exposed hydrophobic segment of other unfolded proteins. Pharmacological chaperones may prohibit protein aggregation by binding and masking hydrophobic parts of unfolded or misfolded proteins. The IBT21 core possesses hydrophobicity comparable to that of 4PBA and TUDCA, which are known to be hydrophobic compounds. This similarity supports the hypothesis that the hydrophobic core of IBT21 may interact with the exposed hydrophobic segments of unfolded or misfolded proteins. Currently, 4PBA and TUDCA are the only two pharmacological chaperones for proteotoxins approved by the FDA for human use. Notably, IBT21 shows much higher chaperone-like activity than 4PBA. In future studies, we will further examine the therapeutic effects of IBTs in model mice with a disease involving ER stress.

In theory, pharmacological chaperones for proteotoxin have one of the following three mechanisms of action: preventing protein aggregation by masking the aggregation-prone region (group I), increasing the folding rate by stabilizing the folding transition state (group II) or decreasing the misfolding rate by stabilizing the native state (group III). We demonstrated that IBT21 directly prevents protein aggregation in an in vitro aggregation inhibition assay, but the second and third mechanisms remain elusive. A recent report suggests that the action of chemical chaperones is determined by the proteins to be folded and the folding environment, as a chemical chaperone that behaves as a group II chaperone for one protein might behave as a group III chaperone for another (*Dandage et al., 2015*). Notably, heat shock response is not influenced by IBT21. It would be difficult to generalize the mechanism of action of pharmacological chaperones based on their effect in one test stress condition or on one test protein. Hence, further characterization of IBT21 appears to require testing on disease-related misfolding-prone proteins and on disease-related stress conditions.

The chemical properties of IBTs that enable them to preferably bind to unfolded or misfolded proteins might be useful for developing diagnostic tools for ER stress-related diseases. Several positron emission tomography (PET) and single photon emission computed tomography (SPECT) tracers for imaging β-amyloid deposits in Alzheimer's disease have been used clinically or in a clinical trial. Alagille et al. designed and synthesized a series of 2-aryl-imidazo[2,1-*b*]benzothiazoles by a combination of the benzothiazole motif and the 2-arylimidazo motif from two potent β-amyloid imaging agents, PIB and IMPY, respectively (*Alagille et al., 2011*). Yousefi et al. demonstrated that [11]C-labelled IBTs have high in vitro and in vivo binding affinities for β-amyloid aggregates (*Alagille et al., 2011*) and that [18]F-labelled IBTs allow high-contrast PET imaging of β-amyloids in

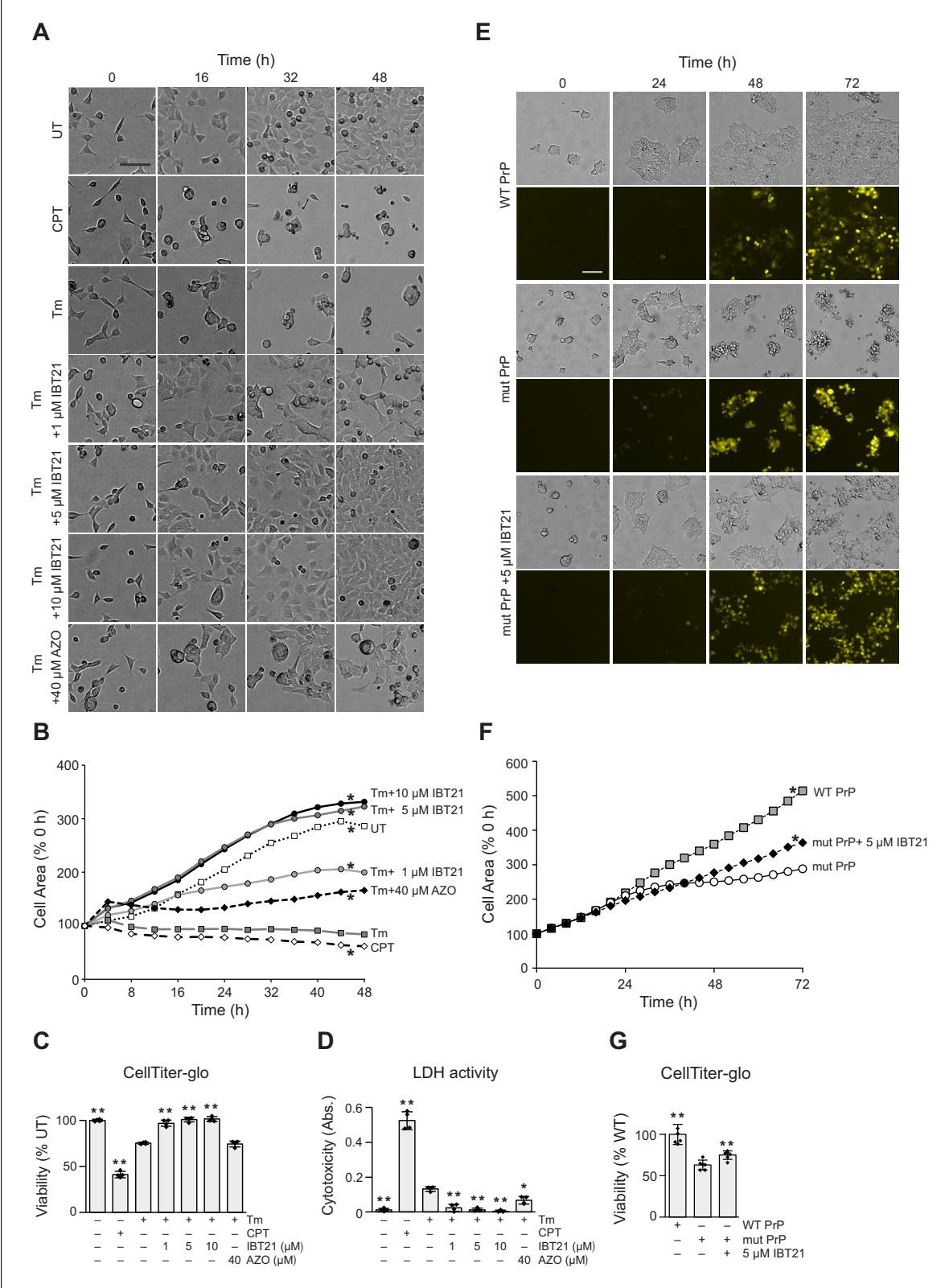

**Figure 7.** Cytoprotective effects of IBT21 during ER stress. (**A**) Representative bright-field images of HEK293A cells treated with 0.2 µg/mL Tm in the presence or absence of compounds for the indicated times. Camptothecin (CPT) was used as a cell death control. The scale bar represents 100 µm. (**B**) Cell growth curve of HEK293A cells treated with 0.2 µg/mL Tm in the presence or absence of compounds for the indicated times. ($n$ = 3); two-way ANOVA with Holm's post hoc test, *p<0.05. (**C–D**) HEK293A cells were treated with 0.2 µg/mL Tm in the presence or absence of compounds for 48 hr.
*Figure 7 continued on next page*

*Figure 7 continued*

Cell viability was determined by an intracellular ATP assay (**C**) or LDH activity (**D**). Error bars show the mean ± SD (*n* = 4). One-way ANOVA with Tukey's post hoc test, *p<0.05 and **p<0.01. (**E**) Representative bright-field and YFP images of HEK293T cells overexpressing YFP-tagged WT or YFP-tagged mutant prion protein (PrP) in the presence or absence of 5 μM IBT21 for the indicated times. The scale bar represents 100 μm. (**F**) Cell growth curve of HEK293T cells overexpressing YFP-tagged WT or YFP-tagged mutant prion protein (PrP) in the presence or absence of 5 μM IBT21 for the indicated times (*n* = 6). Two-way ANOVA with Holm's post hoc test, *p<0.05. (**G**) Intracellular ATP assay of HEK293T cells overexpressing YFP-tagged WT or YFP-tagged mutant prion protein (PrP) in the presence or absence of 5 μM IBT21 for the indicated times. Error bars show the mean ± SD (*n* = 6). One-way ANOVA with Tukey's post hoc test, *p<0.05 and **p<0.01.

The online version of this article includes the following source data for figure 7:

**Source data 1.** Dataset for *Figure 7*.

Alzheimer's disease model mice (*Yousefi et al., 2011*). Together with our results, these findings indicate that IBT21 could be effective for PET and SPECT imaging of ER stress-related diseases.

Targeting improper protein folding and aggregate accumulation, which are the most upstream events in ER stress-related diseases, is an obvious therapeutic strategy with chemical chaperones. It is generally accepted that neurodegenerative disorders such as Alzheimer's disease, Huntington disease and prion disease are caused by the accumulation of neurotoxic protein aggregates, and chemical chaperones have the potential to separate these neurotoxic aggregates. In the case of prion disease, the osmoprotectant TMAO was first applied for preventing PrP$^{SC}$ formation in vitro as a chemical chaperone in the late 1990 s (*Tatzelt et al., 1996*). PrP$^{SC}$ toxicity was reported to cause ER stress by increasing the intracellular calcium released from the ER (*Hetz et al., 2003*). Indeed, a reduction in the levels of the ER molecular chaperone BiP accelerates prion pathogenesis in vivo, indicating that the accumulation of the misfolded form of PrP in the ER is a key event in prion propagation (*Park et al., 2017*). Prion replication has been shown to cause sustained UPR induction with persistent eIF2α phosphorylation (*Moreno et al., 2012*). Repression of protein synthesis diminished the load of unfolded proteins in the ER, but overactivation of eIF2α phosphorylation leads to synaptic failure, spongiosis and neuronal loss. Pharmacological inhibition of PERK by the compound GSK2606414 mitigated neurodegeneration and clinical prion disease in mice (*Moreno et al., 2013*). In this study, we demonstrated that IBT21 presents the most effective treatment effect and that the best treatment regimen may involve a combination of compounds, some of which fine-tune protein synthesis, such as GSK2606414, and others that reduce protein aggregates, such as the chemical chaperone IBT21. More studies with these promising compounds need to be performed to design therapies to treat prion diseases and possibly other neurodegenerative diseases associated with ER stress.

## Materials and methods

### Chemical compounds

A total of 217,765 library compounds were provided by the Drug Discovery Initiative at the University of Tokyo (Tokyo, Japan). 7-Methylsulphonyl-2-phenylimidazo[2,1-*b*]benzothiazole (IBT21) was purchased from Butt Park (Devon, UK). 2-(4-Chlorophenyl)−7-methylsulphonylimidazo[2,1-*b*]benzothiazole (IBT22) was purchased from Specs (Zoetermeer, Netherlands). Azoramide was purchased from Axon Medchem (Groningen, Netherlands). 4-Phenylbutyric acid was purchased from Tokyo Chemical Industry (Tokyo, Japan). Camptothecin was purchased from FUJIFILM Wako Pure Chemical (Osaka, Japan). PERK Inhibitor I (GSK2606414) was purchased from Merck Millipore (MA, USA). IRE1 Inhibitor I (STF083010) was purchased from Toronto Research Chemicals (Ontario, Canada). Site one protease inhibitor (PF429242) was purchased from R and D Systems (MN, USA). Twenty-eight compounds for the SAR study were provided by the Drug Discovery Initiative at the University of Tokyo and quality controlled using an ACQUITY UPLC (Waters, MA, USA) equipped with an ACQUITY UPLC BEH C18 column to determine the purity. The column temperature was 40°C. Mobile phase A consisted of 5.0 mM ammonium acetate in water. Mobile phase B consisted of acetonitrile (ACN). The gradient conditions were as follows: 5% mobile phase B at the time of injection; linear increase to 95% B at 1.00 min; and maintain 95% B until 1.50 min. The flow rate was 0.7 mL/min. The purity measurements were determined by measuring the peak area at 230 nm, and the purity was

calculated based on the observed target peak area as a percentage of the total area. All compounds assayed were $\geq$95% pure. 7-(methylaminocarbonyl)−2-phenylimidazo[2,1-*b*]benzothiazole (IBT29) was synthesized from 4-(methylaminocarbonyl)aniline and phenacyl bromide. More detailed information is shown in the Supplemental information.

## Establishment of reporter cell lines

The HEK293A cell line was provided by Thermo Fisher Scientific (MA, USA) and maintained in DMEM (Nissui Pharmaceutical, Tokyo, Japan) supplemented with 10% (v/v) foetal bovine serum (Thermo Fisher Scientific). The Hap1 cell line was provided by Horizon Genomics (Vienna, Austria) and maintained in IMDM (GE Healthcare, IL, USA) supplemented with 10% (v/v) foetal bovine serum. Cells were cultured at 37°C in a humidified incubator continuously flushed with a mixture of 5% $CO_2$ and 95% air. Cells were authenticated with a morphology check by microscopy and using growth curve analysis with an Operetta CLS instrument (PerkinElmer, MA, USA). All cell lines were confirmed to be mycoplasma-free.

Tandem ER stress response element-2 (ERSE2), unfolded protein response element (UPRE), amino acid response element (AARE) and heat-shock element (HSE) were assembled by Golden Gate cloning with the Multiplex CRISPR/Cas9 Assembly System Kit (Addgene Kit # 1000000055, MA, USA). 10X ERSE2-10X UPRE-5X AARE (EUA), 25X ERSE2, 25X UPRE, 25X AARE and 5X HSE-EGFP reporter vectors were constructed based on the lenti-Cas9-blast vector (gift from Feng Zhang [Addgene plasmid # 52962]). The DNA sequence of each construct was verified on an ABI 3130 DNA sequencer (Thermo Fisher Scientific). HEK293A and Hap1 cells stably expressing the reporter gene were sorted with an S3e Cell Sorter (Bio-Rad, CA, USA).

- ERSE2: GGACGCCGATTGGGCCACGTTGGGAGAGTGCCT
- UPRE: CTCGAGACAGGTGCTGACGTGGCATTC
- AARE: AACATTGCATCATCCCCGC
- HSE: GAACGTTCCCGAA

## Reporter assay

HEK293A-EUA-EGFP, ERSE2-EGFP, UPRE-EGFP and AARE-EGFP cells were seeded onto 96- or 384-well plates (Greiner Bio-One, Kremsmünster, Austria). Tunicamycin (Tm; Merck Millipore) was added in the presence or absence of chemical compounds, followed by incubation overnight. For the dithiothreitol (DTT) experiments, HEK293A-EUA-EGFP cells were treated with 5 mM DTT (FUJI-FILM Wako Pure Chemical) for 4 hr in the presence or absence of 10 µM IBT21. Then, the compounds were washed out, followed by incubation at 37°C overnight. For the heat-shock experiments, HEK293A-HSE-EGFP cells were exposed to hyperthermia in a $CO_2$ incubator at 42°C for 4 hr, followed by incubation at 37°C overnight. Intracellular EGFP fluorescence was measured with Cytation3 (Ex 485 nm, Em 528 nm, BioTek, VT, USA) or Operetta CLS (Ex 475 ± 15 nm, Em 525 ± 25 nm). Cell viability was measured using the alamarBlue Cell Viability Reagent (Thermo Fisher Scientific) with Cytation3 (Ex 560 nm, Em 590 nm). Data analyses were performed using Harmony 4.6 (PerkinElmer) and TIBCO Spotfire Analyst (ver. 7.6.1, TIBCO Software, CA, USA). Structure clustering was performed with ChemFinder ultra (ver. 16.0.1.4, PerkinElmer). The partition coefficient (LogP) was calculated by ChemDraw professional (ver. 16.0.1.4, PerkinElmer). The mean signal-background ratio (S/B) was calculated as (Tm treatment)/(no-ER stress control). The mean Z′ was calculated as 1−(3 [σ Tm + σ DMSO]/[µ Tm−µ DMSO]).

## Immunoblot analysis

Cells were washed in ice-cold PBS and lysed in RIPA buffer containing 50 mM Tris-HCl (pH 7.4), 150 mM NaCl, 2 mM EDTA, 1% (v/v) NP-40% and 0.1% (w/v) SDS with protease inhibitor cocktail (Bimake, TX, USA), phosphatase inhibitor cocktail (Bimake) and 10 µM MG132 (Enzo Life Sciences, NY, USA). The protein concentrations of lysates were measured using the Protein Assay Bicinchoninate Kit (Nacalai Tesque, Kyoto, Japan). Immunoblot analysis was performed as described previously (*Taniuchi et al., 2016*) using Blocking One (Nacalai Tesque) and Immobilon Western HRP (Merck Millipore). Protein was visualized using Ez-Capture II (ATTO, Tokyo, Japan), and the signal intensities were quantified using the ImageJ software program (version 1.51) (*Schneider et al., 2012*). The antibodies for immunoblotting were as follows: anti-ATF4 (10835–1-AP, Proteintech, IL, USA), anti-

SEL1L (ab78298, Abcam, Cambridge, UK), anti-HERPUD1 (26730S, Cell Signalling Technology, MA, USA), anti-DNAJB9 (13157–1-AP, Proteintech), anti-4E-BP1 (9644S, Cell Signalling Technology), anti-ATF6α (73–505, BioAcademia, Osaka, Japan), anti-β-actin (PM053, Medical and Biological Laboratories, Aichi, Japan) and anti-KDEL (M181-3, Medical and Biological Laboratories) antibodies. Anti-CHOP and anti-GADD34 antibodies were kindly gifted by Dr. David Ron (Cambridge Institute for Medical Research, Cambridge, UK).

## Overexpression of UPR transcription factors

The pcDNA3.1-human ATF4 plasmid was introduced into the HEK293T cell line (ATCC, VA, USA) by transfection with polyethylenimine (Polysciences, PA, USA), followed by incubation at 37°C for 24 hr, after which the cells were incubated in the presence or absence of 10 μM IBT21 for 24 hr. The pcDNA3.1-human ATF6 (N) plasmid was introduced into the HEK293A cell line by transfection with polyethylenimine, followed by incubation at 37°C for 24 hr, after which the cells were incubated in the presence or absence of 10 μM IBT21 for 12 or 24 hr.

## Nonradioactive measurements of protein synthesis

To measure protein synthesis, we used the nonradioactive surface sensing of translation (SUNsET) technique. Briefly, 293A-EUA-EGFP cells were pre-treated with 10 μg/mL cycloheximide (Nacalai Tesque) or 10 μM IBT21 for 1–4 hr, followed by 2 μg/mL puromycin (GoldBio, MO, USA) treatment for 30 min. Total protein was extracted and immunoblot analysis was performed with an anti-puromycin antibody (EQ0001, Kerafast, MA, USA). Ponceau S (Sigma-Aldrich, MO, USA) was used as a loading control.

## Aggregated protein detection

An aggregated protein detection assay was performed with the ProteoStat Aggresome Detection Kit (Enzo Life Sciences) according to the manufacturer's instructions. For cell analyses, the specimens were viewed with a BZ-X700 device (Ex 545 ± 12.5 nm, Em 605 ± 35 nm, Keyence, Osaka, Japan) and Operetta CLS (Ex 475 ± 15 nm, Em 610 ± 40 nm). For in vitro analysis, 5 mg/mL human recombinant insulin (FUJIFILM Wako Pure Chemical) was incubated at 37°C overnight with 100 mM DTT and 10% (v/v) DMSO in the presence or absence of IBT21, azoramide or 4PBA. Insulin aggregation was quantified with an EnVision2102 (Ex 480 nm, Em 595 nm, PerkinElmer).

## High molecular weight detergent-resistant complex detection

High molecular weight detergent-resistant complex detection was performed as described previously (Hisanaga et al., 2018). Briefly, cultured cells were incubated for 5 min in ice-cold PBS containing 20 mM N-ethylmaleimide (Sigma-Aldrich) and then collected by scraping in lysis buffer (0.5% Triton X-100, 20 mM HEPES [pH 7.4], 250 mM sucrose, 100 mM NaCl and 2.5 mM $CaCl_2$) with protease inhibitors. Equal amounts of protein were adjusted to 0.8% SDS, layered upon separation buffer (0.5% Triton X-100, 0.8% SDS, 20 mM HEPES [pH 7.4], and 20% glycerol) and centrifuged at 100,000 × g for 55 min with an Optima Ultracentrifuge (Beckman Coulter, CA, USA). For immunoblot analysis, the resulting pellet was dissolved in urea loading buffer (9.6 M urea, 1.36% SDS, 40 mM Tris [pH 6.8], 12% glycerol, 100 mM DTT and 0.002% [w/v] bromophenol blue), boiled for 5 min and then subjected to SDS-PAGE. For mass spectrometry analysis, the pellet was rinsed with ice-cold 50 mM ammonium bicarbonate (Sigma-Aldrich) and centrifuged at 100,000 × g for 80 min. Proteins in the pellet were dissolved in 8 M urea and 50 mM Tris (pH 8.0), reduced, alkylated, diluted to 1 M urea and digested with trypsin/Lys-C mix (Promega, WI, USA) at 37°C for 16 hr. The digests were desalted using GL-Tip SDB (GL Sciences, Tokyo, Japan). LC-MS/MS analysis of the resultant peptides was carried out on an EASY-nLC 1200 UHPLC connected to a Q Exactive Plus mass spectrometer through a nanoelectrospray ion source (Thermo Fisher Scientific). The peptides were separated on a 75 μm inner diameter ×150 mm C18 reversed-phase column (Nikkyo Technos, Tokyo, Japan) with a linear gradient ranging from 4–28% ACN for min 0–100, followed by an increase to 80% ACN for min 100–110. The mass spectrometer was operated in data-dependent acquisition mode with a top 10 MS/MS method. MS1 spectra were measured with a resolution of 70,000, an automatic gain control (AGC) target of $1 \times 10^6$ and a mass range from 350 to 1,500 $m/z$. MS/MS spectra were triggered at a resolution of 17,500, an AGC target of $5 \times 10^4$, an isolation window of

2.0 *m/z*, a maximum injection time of 60 ms and a normalized collision energy of 27. Dynamic exclusion was set to 10 s. Raw data were directly analyzed against the SwissProt database restricted to *H. sapiens* using Proteome Discoverer version 2.2 (Thermo Fisher Scientific) for identification and label-free precursor ion quantification. GO enrichment was performed with Genomatix Genome Analyzer (Genomatix, München, Germany).

## Synthesis of the diazido-IBT21 probe

IBT21 and diazido-IBT21 were synthesized from 6-(methylsulphonyl)benzo[*d*]thiazol-2-amine and phenacyl bromide or 3-azido-5-(azidomethyl)phenacyl bromide, respectively. More detailed information is shown in the Supplemental information.

## Identification of proteins interacting with the diazido-IBT21 probe

HEK293A cells were treated with 0.2 µg/mL Tm for 18 hr, followed by the addition of 1 µM diazido-IBT21 probe in the presence or absence of 10 µM IBT22 (competitor) for 3 hr at 37°C. Photo-irradiation was performed at room temperature for 180 s using UVG-54 (254 nm, 6 W, from a distance of 1 cm, UVP, CA USA). Cell lysates were prepared in 20 mM HEPES (pH 7.4), 150 mM NaCl, 1.5 mM MgCl$_2$, 1% NP-40, 2 M urea and protease inhibitors on ice. For each sample, 800 µg of lysate was reacted with the Click-iT Protein Reaction Buffer Kit (Thermo Fisher Scientific) and biotin-alkyne (PEG4 carboxamide-Propargyl Biotin, Thermo Fisher Scientific) according to the manufacturer's instructions. Labelled proteins were precipitated with Streptavidin Sepharose High Performance (GE Healthcare), resuspended in 50 mM ammonium bicarbonate and digested with trypsin/Lys-C mix for 16 hr at 37°C. The digests were desalted using GL-Tip SDB. LC-MS/MS analysis of the resultant peptides was carried out on a Dionex UltiMate 3000 UHPLC system connected to an Orbitrap Fusion mass spectrometer through a nanoelectrospray ion source (Thermo Fisher Scientific). The peptides were separated on a 75 µm inner diameter ×120 mm C18 reversed-phase column with a linear gradient ranging from 4–40% ACN for min 10–110, followed by an increase to 80% ACN for min 110–120. The mass spectrometer was operated in data-dependent acquisition mode with a maximum duty cycle of 3 s. MS1 spectra were measured with a resolution of 120,000, an AGC target of $4 \times 10^5$ and a mass range from 375 to 1,500 *m/z*. HCD MS/MS spectra were acquired at a resolution of 15,000, an AGC target of $5 \times 10^4$, an isolation window of 4.0 *m/z*, a maximum injection time of 54 ms and a normalized collision energy of 30. Dynamic exclusion was set to 10 s. Raw data were directly analyzed against the SwissProt database restricted to *H. sapiens* using Proteome Discoverer version 2.2 for identification and label-free precursor ion quantification.

## Viability assay

HEK293A cells were seeded on a 384-well plate. Tunicamycin was added in the presence or absence of compounds and then incubated for 48 hr. Cell viability was measured using CellTiter-Glo (Promega) and Cytotoxicity LDH Assay Kit-WST (Abs 490 nm, FUJIFILM Wako Pure Chemical) with Cytation3. Cell images were viewed with Operetta CLS and analyzed with Harmony 4.6.

## Overexpression of mutant prion protein

The mutant PrP (YFP-PrP C179A/S232W) expression vector was kindly gifted by Dr. Ramanujan S. Hegde (MRC Laboratory of Molecular Biology, Cambridge, UK). Each plasmid was introduced into the HEK293T cell line by transfection with polyethylenimine in the presence or absence of 5 µM IBT21. Cell images were viewed with Operetta CLS and analysed with Harmony 4.6.

## Statistical analysis

The data shown in each figure are expressed as the means ± standard deviations. The statistical analyses were performed using the StatFlex software (Ver. 6.0, Artech, Osaka, Japan). In comparisons of 3 or more groups, we used a one-way ANOVA with Tukey's post hoc test or two-way ANOVA with the Holm's test. A p value less than 0.05 was considered significant.

# Synthesis of 7-methylsulphonyl-2-phenylimidazo[2,1-*b*]benzothiazole (IBT21) and 7-methylsulphonyl-2-(3-azido-5-(azidomethyl)phenyl) imidazo[2,1-*b*] benzothiazole (diazido-IBT21) photoaffinity labelling probes

All reactions were performed with dry glassware under an argon atmosphere unless otherwise noted. Analytical thin-layer chromatography (TLC) was performed on precoated (0.25 mm) silica gel plates (Silica Gel 60 F254, Merck Chemicals). Column chromatography was conducted using Biotage SNAP Ultra 50 g with medium pressure liquid chromatography (W-Prep 2XY A-type, Yamazen, Osaka, Japan). Preparative TLC was performed on silica gel (Wakogel B-5F, FUJIFILM Wako Pure Chemical). Melting point (mp) values were measured on an OptiMelt MPA100 automated melting point apparatus (Stanford Research Systems, CA, USA) and are uncorrected. Infrared (IR) spectra were measured by the diffuse reflectance method on an IRPrestige-21 spectrometer (Shimadzu, Kyoto, Japan) attached to a DRS-8000A accessory with the absorption band given in cm$^{-1}$. $^1$H NMR spectra were measured by a Bruker AVANCE 500 spectrometer or a Bruker AVANCE 400 spectrometer (Bruker, MA, USA) at 500 or 400 MHz, respectively. $^{13}$C NMR spectra were measured by a Bruker AVANCE 500 spectrometer at 126 MHz. CDCl$_3$ (Kanto Chemical, Tokyo, Japan) or DMSO-$d_6$ (Cambridge Isotope Laboratories, MA, USA) was used as a solvent for obtaining NMR spectra. Chemical shifts (δ) are given in parts per million (ppm) downfield from (CH$_3$)$_4$Si (δ 0.00 for $^1$H NMR in CDCl$_3$) or the solvent peak (δ 77.0 for $^{13}$C NMR in CDCl$_3$, δ 2.49 for $^1$H NMR and δ 39.5 for $^{13}$C NMR in DMSO-$d_6$) as an internal reference with coupling constants (J) in hertz (Hz). The abbreviations s, d, and br signify singlet, doublet, and broad, respectively.

## Step 1 (*Scheme 1*)

**Scheme 1.** Synthesis of compounds S1 - S3.

3-Azido-5-(azidomethyl)benzoic acid (**S1**) was prepared according to the previous report (*Yoshida et al., 2014*). A solution of oxalyl chloride (2.05 g, 16.2 mmol) in dichloromethane (6.0 mL) at 0˚C was added to a solution of 3-azido-5-(azidomethyl)benzoic acid (**S1**) (1.75 g, 8.02 mmol) and *N,N*-dimethylformamide (DMF) (59.7 mg, 0.817 mmol) in dichloromethane (6.0 mL). After stirring for 3 hr at room temperature, the mixture was concentrated under reduced pressure. The residue was dissolved in a mixed solvent of acetonitrile (MeCN) (5.0 mL) and tetrahydrofuran (THF) (5.0 mL), and a solution of trimethylsilyl diazomethane (TMSCHN$_2$) in diethyl ether (2.0 M, 16.0 mL, 32 mmol) at 0˚ C was added to the solution. After stirring for 17 hr at room temperature, the mixture was filtered through a plug of cotton, and the filtrate was concentrated under reduced pressure. The residue was purified by flash column chromatography (*n*-hexane/dichloromethane = 76/24 to 55/45) to give 3'-azido-5'-(azidomethyl)-α-diazoacetophenone (**S2**) (1.01 g, 4.17 mmol, 52.0% from **S1**) as a yellow solid. Mp 57–58˚C; TLC $R_f$ 0.21 (*n*-hexane/ethyl acetate = 5/1); $^1$H NMR (CDCl$_3$, 500 MHz) δ 4.42 (s, 2H), 5.91 (s, 1H), 7.13–7.15 (m, 1H), 7.38–7.40 (m, 1H), 7.43–7.45 (m, 1H); $^{13}$C NMR (CDCl$_3$, 126 MHz) δ 53.9 (1C), 54.9 (1C), 117.0 (1C), 122.1 (1C), 122.3 (1C), 138.2 (1C), 138.7 (1C), 141.6 (1C), 184.7 (1C); IR (KBr, cm$^{-1}$) 740, 1177, 1236, 1307, 1366, 1442, 1589, 2107; Anal. calcd. for C$_9$H$_6$N$_8$O: C, 44.63; H, 2.50; N, 46.27%; Found: C, 44.71; H, 2.49; N, 46.33%.

Aqueous hydrobromic acid (48%, 0.57 mL, 5.0 mmol) at 0˚C was added to a solution of 3'-azido-5'-(azidomethyl)-α-diazoacetophenone (**S2**) (401 mg, 1.66 mmol) in acetic acid (AcOH) (2.5 mL). After stirring for 70 min at room temperature, saturated aqueous sodium bicarbonate was added to the mixture to adjust the pH to 8. The mixture was extracted with ethyl acetate (20 mL ×3), and the combined organic extract was washed with brine (20 mL) and dried (Na$_2$SO$_4$). After filtration, the

filtrate was concentrated under reduced pressure. The residue was purified by flash column chromatography (*n*-hexane/ethyl acetate = 92/8) to give 3-azido-5-(azidomethyl)phenacyl bromide (**S3**) (413 mg, 1.40 mmol, 84.5%) as a brown oil. TLC $R_f$ 0.51 (*n*-hexane/ethyl acetate = 5/1); $^1$H NMR (CDCl$_3$, 500 MHz) δ 4.44 (s, 2H), 4.46 (s, 2H), 7.20–7.23 (m, 1H), 7.57–7.59 (m, 1H), 7.66–7.68 (m, 1H); $^{13}$C NMR (CDCl$_3$, 126 MHz) δ 30.5 (1C), 53.7 (1C), 118.8 (1C), 123.3 (1C), 124.4 (1C), 135.8 (1C), 138.5 (1C), 141.8 (1C), 190.1 (1C); IR (KBr, cm$^{-1}$) 851, 1261, 1312, 1347, 1443, 1592, 1688, 2108; HRMS (ESI$^+$) *m/z* 316.9760 ([M+Na]$^+$, C$_9$H$_7$$^{79}$BrN$_6$NaO$^+$ requires 316.9757).

## Step 2 (*Scheme 2*)

**Scheme 2.** Synthesis of compounds S4 - IBT21 or diazido-IBT21.

Bromine (0.26 mL, 5.04 mmol) at 0°C was slowly added to a solution of 4-(methylsulphonyl)aniline (**S4**) (858 mg, 5.01 mmol) and potassium isothiocyanate (2.43 g, 25.0 mmol) in acetic acid (15 mL). After stirring for 3 days with gradual warming to room temperature, saturated aqueous sodium bicarbonate (100 mL) and sodium bicarbonate (25 g) were added to the mixture at 0°C to neutralize the mixture. After filtration of the mixture and washing with water (150 mL), the filtrate was extracted with ethyl acetate (5 mL ×10). The combined organic extract was concentrated under reduced pressure to give 6-(methylsulphonyl)benzo[*d*]thiazol-2-amine (**S5**) (45.4 mg, 0.198 mmol, 58.2%) as a yellow solid. Mp 210–212°C; TLC $R_f$ 0.17 (dichloromethane/methanol = 30/1); $^1$H NMR (DMSO-$d_6$, 500 MHz) δ 3.17 (s, 3H), 7.47 (d, *J* = 8.0 Hz, 1H), 7.71 (dd, *J* = 8.0, 2.0 Hz, 1H), 8.00 (br s, 2H), 7.71 (d, *J* = 2.0 Hz, 1H); $^{13}$C NMR (DMSO-$d_6$, 126 MHz) δ 44.2 (1C), 117.3 (1C), 120.7 (1C), 124.8 (1C), 131.5 (1C), 132.4 (1C), 156.8 (1C), 170.2 (1C); IR (KBr, cm$^{-1}$) 968, 1140, 1258, 1452, 1529, 1643, 3067, 3399; HRMS (ESI$^+$) *m/z* 250.9912 ([M+Na]$^+$, C$_8$H$_8$N$_2$NaO$_2$S$_2$$^+$ requires 250.9919).

A suspension of 6-(methylsulphonyl)benzo[*d*]thiazol-2-amine (**S5**) (45.3 mg, 0.198 mmol) and phenacyl bromide (**S6**) (47.6 mg, 0.239 mmol) in 2-methoxyethanol (0.75 mL) was refluxed (bath temperature: 135°C) for 3.5 hr. Aqueous ammonia (10%, 2 mL) at 0°C was added to the mixture. After stirring for 15 min at room temperature and filtration, the precipitate was collected and washed with water (20 mL) and cold ethanol (1 mL). The collected solid was dried under reduced pressure. The crude product was purified by preparative TLC (dichloromethane/methanol = 50/1) and reprecipitated with ethyl acetate and dichloromethane to give 7-methylsulphonyl-2-phenylimidazo[2,1-*b*]benzothiazole (IBT21) (9.9 mg, 34 μmol, 17%) as a pale yellow solid. Mp 245–247°C; TLC $R_f$ 0.64 (dichloromethane/methanol = 30/1); $^1$H NMR (CDCl$_3$, 500 MHz) δ 3.14 (s, 3H), 7.31–7.37 (AA'BB'C, 1H), 7.42–7.48 (AA'BB'C, 2H), 7.71 (d, *J* = 8.4 Hz, 1H), 7.86–7.90 (AA'BB'C, 2H), 8.04 (s, 1H), 8.06 (dd, *J* = 8.4, 2.0 Hz, 1H), 8.34 (d, *J* = 2.0 Hz, 1H); $^{13}$C NMR (CDCl$_3$, 126 MHz) δ 44.9 (1C), 107.1 (1C), 113.0 (1C), 124.3 (1C), 125.4 (2C), 125.8 (1C), 128.1 (1C), 128.9 (2C), 131.6 (1C), 133.1 (1C), 135.4 (1C), 137.0 (1C), 148.8 (1C), 149.0 (1C); IR (KBr, cm$^{-1}$) 800, 951, 1096, 1145, 1209, 1273, 1298, 1312, 1503; HRMS (ESI$^+$) *m/z* 351.0230 ([M+Na]$^+$, C$_{16}$H$_{12}$N$_2$NaO$_2$S$_2$$^+$ requires 351.0230).

A suspension of 6-(methylsulphonyl)benzo[*d*]thiazol-2-amine (**S5**) (22.8 mg, 0.100 mmol) and 3-azido-5-(azidomethyl)phenacyl bromide (**S3**) (35.4 mg, 0.120 mmol) in 2-methoxyethanol (0.37 mL) was refluxed (bath temperature: 135°C) for 1.5 hr. Aqueous ammonia (2%, 6 mL) at 0°C was added to the mixture. After stirring for 15 min at room temperature and filtration, the precipitate was collected and washed with water (30 mL). The collected solid was dried under reduced pressure and then dissolved in dichloromethane (20 mL). After filtration of the mixture, the filtrate was concentrated to afford a brown solid involving a small amount of impurity. The crude solid was purified by preparative TLC (dichloromethane/methanol = 70/1) to give 7-methylsulphonyl-2-(3-azido-5-(azidomethyl)phenyl)imidazo[2,1-*b*]benzothiazole (diazido-IBT21) (4.6 mg, 11 μmol, 11%) as a pale yellow solid. Mp 207°C (decomp.); TLC $R_f$ 0.55 (dichloromethane/methanol = 30/1); $^1$H NMR (CDCl$_3$, 500 MHz) δ 3.14 (s, 3H), 4.42 (s, 2H), 6.95 (dd, $J$ = 1.8, 1.8 Hz, 1H), 7.53 (dd, $J$ = 1.8, 1.8 Hz, 1H), 7.62 (dd, $J$ = 1.8, 1.8 Hz, 1H), 7.81 (d, $J$ = 8.4 Hz, 1H), 8.08 (dd, $J$ = 8.4, 1.6 Hz, 1H), 8.10 (s, 1H), 8.36 (d, $J$ = 1.6 Hz, 1H); $^{13}$C NMR (CDCl$_3$, 126 MHz) δ 44.9 (1C), 54.3 (1C), 108.0 (1C), 113.1 (1C), 115.6 (1C), 118.0 (1C), 121.4 (1C), 124.4 (1C), 126.0 (1C), 131.6 (1C), 135.2 (1C), 135.5 (1C), 137.4 (1C), 138.1 (1C), 141.4 (1C), 147.4 (1C), 149.1 (1C); IR (KBr, cm$^{-1}$) 750, 1150, 1303, 1501, 2110; HRMS (ESI$^+$) *m/z* 447.0402 ([M+Na]$^+$, C$_{17}$H$_{12}$N$_8$NaO$_2$S$_2^+$ requires 447.0417).

## Acknowledgements

We thank C Kimura (Tokushima University) for help with manuscript preparation, J Kakegawa (Japan Tobacco Inc) for technical assistant, Joint Usage and Joint Research Programs, the Institute of Advanced Medical Sciences, Tokushima University and the Drug Discovery Initiative at the University of Tokyo for the chemical libraries.

## Additional information

### Funding

| Funder | Grant reference number | Author |
| --- | --- | --- |
| Japan Agency for Medical Research and Development | JP18nk0101336 | Seiichi Oyadomari |
| Japan Agency for Medical Research and Development | JP17am0101086 | Hirotatsu Kojima |
| Japan Society for the Promotion of Science | 16H05222 (Grant-in-Aid for Scientific Research) | Seiichi Oyadomari |
| Japan Agency for Medical Research and Development | JP18am0101098 | Takamitsu Hosoya |
| Japan Agency for Medical Research and Development | JP18am0101098 | Takamitsu Hosoya |

The funders had no role in study design, data collection and interpretation, or the decision to submit the work for publication.

### Author contributions

Keisuke Kitakaze, Eri Kawano, Yoshimasa Hamada, Tomoko Kuribara, Investigation; Shusuke Taniuchi, Resources, Methodology; Masato Miyake, Supervision, Methodology; Miho Oyadomari, Suguru Yoshida, Investigation, Methodology; Hirotatsu Kojima, Resources, Data curation, Funding acquisition, Validation, Methodology; Hidetaka Kosako, Formal analysis, Investigation; Takamitsu Hosoya, Supervision, Funding acquisition, Validation, Methodology; Seiichi Oyadomari, Conceptualization, Supervision, Funding acquisition, Methodology, Project administration

### Author ORCIDs

Keisuke Kitakaze http://orcid.org/0000-0002-4852-1257
Suguru Yoshida http://orcid.org/0000-0001-5888-9330
Seiichi Oyadomari https://orcid.org/0000-0001-6766-1485

Decision letter and Author response

Decision letter https://doi.org/10.7554/eLife.43302.sa1

Author response https://doi.org/10.7554/eLife.43302.sa2

## Additional files

### Supplementary files

• Transparent reporting form

### Data availability

All data generated or analysed during this study are included in the manuscript and supporting files.

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
