## [Decision Letter]

**Acceptance summary:**

In this work, Kitakaze et al. apply cell-based high throughput screening and extensive mechanistic follow-up to identify a new class of chemical chaperones based on the imidazo[2,1-*b*]benzothiazole scaffold that can specifically address ER proteostasis defects, providing a new type of tool to modulate proteostasis that has multiple advantages over other known chemical chaperones, including applicability at lower concentrations. The work will be relevant to laboratories studying protein misfolding and aggregation diseases, as well as those with general interests in proteostasis and stress responses.

**Decision letter after peer review:**

Thank you for submitting your article "Cell-based HTS identifies a chemical chaperone for preventing ER protein aggregation and proteotoxicity" for consideration by *eLife*. Your article has been reviewed by Vivek Malhotra as the Senior Editor, a Reviewing Editor, and three reviewers. The following individuals involved in review of your submission have agreed to reveal their identity: Claudio Hetz (Reviewer #3).

The reviewers have discussed the reviews with one another and the Reviewing Editor has drafted this decision to help you prepare a revised submission.

Summary:

In this manuscript, Kitakaze et al., use cell-based high throughput screening to identify new compounds that can mitigate tunicamycin (Tm)-induced ER stress. Using this approach, they identify a series of compounds containing the IBT substructure. They go on to demonstrate that their compounds have improved potency compared to other UPR modulators/chemical chaperones and that administration of their compounds attenuates UPR signaling in response to Tm-induced ER stress. Furthermore, they demonstrate that administration of their compound IBT21 attenuates Tm-induced protein aggregation in cells and DTT induced protein aggregation in vitro. Similarly, IBT21 improved cellular stress-resistance to two different proteotoxic insults (Tm and prion protein aggregation). These results are consistent with IBTs acting as chemical chaperones. Lastly, they use an MS/photocrosslinking approach to discern potential targets of IBT21, identifying numerous aggregation-prone proteins. Together, these results suggest that IBT21 functions as a chemical chaperone to attenuate ER stress.

Essential revisions:

1) A major issue with the authors' central conclusion is that IBT21 may influence many of the observed readouts by reducing transcription/translation, rather than by functioning as a chemical chaperone. The authors primarily use Tm to induce ER stress in this manuscript. Tm requires new protein synthesis to induce ER stress. Thus, if IBT21 (or other compounds) reduce transcription or translation the stress would also be mitigated. The authors must conclusively address this possibility before the conclusion that IBT21 functions as a chemical chaperone can be properly evaluated. Three key experiments that are required are: (1) to overexpress an active UPR transcription factor and show that the compound does not affect the ability to induce UPR target genes; (2) demonstrate that IBT21 does not influence translation during ER stress using [35S] metabolic labeling or the like; (3) show that IBT21 can also reduce ER stress caused by Tg or DTT, two compounds that do not require protein translation to induce ER protein misfolding/aggregation. It is possible that the chemical chaperoning mechanism is correct, but especially considering that these compounds have a planar conjugated ring system similar to many DNA intercalators, it is important to really demonstrate that the observed effects cannot be explained by a simpler mechanism.

2) Similarly, in the experiment evaluating levels of BiP in the aggregated fraction (Figure 5E), the authors must evaluate whether compound treatment decreases total BiP to rule out changes in expression or degradation of BiP leading to the change in the aggregated fraction.

3) Is IBT21 chemical chaperone activity restricted to ER proteins? It seems unlikely that it would be, if the chemical chaperoning mechanism is correct. The authors should assess IBT21 effects on protein misfolding induced by heat shock or other cytosolic stressors. Does IBT21 suppress induction of heat shock factor I target genes induced by heat or proteotoxic compounds?

4) The azoramide positive control is not working in Figure 4D. This experiment should be repeated using alternative compounds, such as those employed in Figure 1—figure supplement 1.

5) Another major issue is that the described SAR trends are not well-supported by the data. A key and important example is the claim that a 'short substituent endowed the scaffold with the desired activity.' All compounds that contain an extended R2 substituent have an amide bond and it is unclear from SAR whether the amide bond is the issue or the long chain. To justify such a claim, a control with a methyl-substituted amide is needed or, alternatively, the authors could append longer chains onto a sulfonyl or ester. The authors should also moderate other SAR claims as needed based on the data.

6) From the ten inhibitor hits chosen, the authors mention they reduce it to four with a shared scaffold, but they never mention if the other six have higher or lower biological activities, nor do they disclose the structures in the manuscript. This information should be provided, along with further discussion regarding why they were discarded.

7) In the MS experiments conducted to identify all proteins in the aggregated fraction and those that are IBT21-sensitive (Figure 5G), reproducibility needs to be addressed in independent MS experiments and/or by immunoblotting of selected proteins. For the corresponding GO analysis, the authors need to show that the GO results are different for the entire 1200 proteins compared to just IBT-sensitive for the results to be meaningful.

8) In the MS experiment shown in Figure 6E that is measuring the ability of IBT22 to block labeling of proteins by the photocrosslinking probe, the data presentation makes it difficult to determine which targets are the most significantly blocked from labeling and by how many fold compared to DMSO. It would be helpful if the authors would adopt a more standard presentation of the data such as that shown in Lanning et al., 2014). The authors must clarify what threshold (fold-change) is being used to determine a protein is a bona fide target versus nonspecific target. Furthermore, how many total proteins from Figure 6E overlap with those in the IBT-sensitive fraction from Figure 5G?

9) Finally, the authors must assess whether the IBT compound series functions similarly to ameliorate ER stress in other cell lines – preferably primary cells.

[Editors' note: further revisions were requested prior to acceptance, as described below.]

Thank you for resubmitting your work entitled "Cell-based HTS Identifies a chemical chaperone for preventing ER protein aggregation and proteotoxicity" for further consideration at *eLife*. Your revised article has been favorably evaluated by Vivek Malhotra (Senior Editor), a Reviewing Editor, and two reviewers.

The manuscript has been improved but there are some remaining issues that need to be addressed before acceptance, as outlined below:

Essential Revisions:

1) New experiments are only included as supplementary figures. Selected key data, and instances in which experiments were repeated with more appropriate controls, should be incorporated into the main figures.

2) In the Discussion section, please add a section addressing the surprising finding that the heat shock response is not influenced by their compounds.

3) There are 58 overlapped proteins for the Competed and Selective (1205 proteins) and IBT-sensitive (125 proteins) groups. However, only two are highlighted and they are not among the strongest hits from either group. Can anything be learned from overlapping stronger hits? Does this data suggest IBT has over 1000 off-targets (a lot) or is the conclusion that it is stabilizing many more proteins that are not detected in the experiment that determined IBT-sensitive compounds?

4) For the SAR discussion, it would be useful to highlight what is preferred and what is not preferred as opposed to just what is preferred.

---

## [Author Response]

Essential revisions:1) A major issue with the authors' central conclusion is that IBT21 may influence many of the observed readouts by reducing transcription/translation, rather than by functioning as a chemical chaperone. The authors primarily use Tm to induce ER stress in this manuscript. Tm requires new protein synthesis to induce ER stress. Thus, if IBT21 (or other compounds) reduce transcription or translation the stress would also be mitigated. The authors must conclusively address this possibility before the conclusion that IBT21 functions as a chemical chaperone can be properly evaluated. Three key experiments that are required are: (1) to overexpress an active UPR transcription factor and show that the compound does not affect the ability to induce UPR target genes; (2) demonstrate that IBT21 does not influence translation during ER stress using [35S] metabolic labeling or the like; (3) show that IBT21 can also reduce ER stress caused by Tg or DTT, two compounds that do not require protein translation to induce ER protein misfolding/aggregation. It is possible that the chemical chaperoning mechanism is correct, but especially considering that these compounds have a planar conjugated ring system similar to many DNA intercalators, it is important to really demonstrate that the observed effects cannot be explained by a simpler mechanism.

We agree that further experimental evidence is needed to rule out the possibility that IBT21 mitigates ER stress by reducing transcription or translation in order to provide a more compelling conclusion. We thank the reviewers for suggesting these three important experiments and now show data to support the chemical chaperone function of IBT21 as follows:

1-(1) We demonstrated that IBT21 does not reduce transcriptional UPR induction. Namely, IBT21 did not change the expression of CHOP and 4E-BP1, downstream targets of ATF4, in ATF4-overexpressing cells. Additionally, IBT21 did not change the expression of BiP in cells overexpressing an active form of ATF6. The results of this experiment are shown in subsection “Inhibitory effects of IBT21 on activation of the three UPR branches under ER stress conditions”, and in Figure 4—figure supplement2.

1-(2) We noted that IBT21 does not affect protein translation using puromycin-labelled proteins. The results of this experiment are shown in subsection “Inhibitory effects of IBT21 on activation of the three UPR branches under ER stress conditions”, and in Figure 4—figure supplement 3.

1-(3) We showed that IBT21 reduced DTT-induced ER stress by monitoring the activation of EUA-EGFP reporter cells. The results of this experiment are shown in subsection “Inhibitory effects of IBT21 on activation of the three UPR branches under ER stress conditions”, and in Figure 4—figure supplement 4.

2) Similarly, in the experiment evaluating levels of BiP in the aggregated fraction (Figure 5E), the authors must evaluate whether compound treatment decreases total BiP to rule out changes in expression or degradation of BiP leading to the change in the aggregated fraction.

We have included the levels of BiP in the INPUT fraction (Figure 5E-F).To rule out the IBT21 effect on reduced expression or enhanced degradation of BiP, we have added data showing that IBT21 did not change BiP induction in the cells overexpressing an active form of ATF6. The results of this experiment are shown in subsection “In vivoeffects of IBT21 on protein aggregation during ER stress”, and in Figure 5E-F and Figure 4—figure supplement 2.

3) Is IBT21 chemical chaperone activity restricted to ER proteins? It seems unlikely that it would be, if the chemical chaperoning mechanism is correct. The authors should assess IBT21 effects on protein misfolding induced by heat shock or other cytosolic stressors. Does IBT21 suppress induction of heat shock factor I target genes induced by heat or proteotoxic compounds?

Thank you for raising this point; we too had initially assumed that IBT21 suppresses protein misfolding under heat shock conditions as a chemical chaperone. Unfortunately, IBT21 could not suppress the induction of the heat shock response by heat shock in our hands. One might expect that IBT21 preferably prevents protein aggregation in the ER rather than in the cytosol. However, these experiments are challenging because the results could be varied depending upon the stress conditions. This would be a significant undertaking and is out of scope for this initial study. The results of this experiment are shown on subsection “Inhibitory effects of IBT21 on activation of the three UPR branches under ER stress conditions”, and in Figure 4—figure supplement 5.

4) The azoramide positive control is not working in Figure 4D. This experiment should be repeated using alternative compounds, such as those employed in Figure 1—figure supplement 1.

We repeated the experiment with a PERK inhibitor, an S1P inhibitor and an IRE1 inhibitor as positive controls. The results of this experiment are shown in Figure 4—figure supplement 1.

5) Another major issue is that the described SAR trends are not well-supported by the data. A key and important example is the claim that a 'short substituent endowed the scaffold with the desired activity.' All compounds that contain an extended R2 substituent have an amide bond and it is unclear from SAR whether the amide bond is the issue or the long chain. To justify such a claim, a control with a methyl-substituted amide is needed or, alternatively, the authors could append longer chains onto a sulfonyl or ester. The authors should also moderate other SAR claims as needed based on the data.

We newly synthesized IBT29 with a methyl-substituted amide at R2. The stronger activity of IBT29 than IBT21 justifies our claim. The results of this experiment are shown in subsection “Structure-activity relationship study of imidazo[2,1-*b*]benzothiazole derivatives (IBTs)”, and in Figure 2—figure supplement 2. The chemical structure required for the activity has been incorporated into the Discussion section, and the method for IBT29 synthesis is described in the subsection “Chemical compounds”.

6) From the ten inhibitor hits chosen, the authors mention they reduce it to four with a shared scaffold, but they never mention if the other six have higher or lower biological activities, nor do they disclose the structures in the manuscript. This information should be provided, along with further discussion regarding why they were discarded.

The remaining six compounds were all singletons with no common chemical structure. Therefore, the analyses of the six compounds were prioritized lower, and further investigation is needed for publication. We would like to address the results in a future study. We have incorporated this explanation into the Results section subsection “Cell-based high-throughput screen for chemical chaperones”.

7) In the MS experiments conducted to identify all proteins in the aggregated fraction and those that are IBT21-sensitive (Figure 5G), reproducibility needs to be addressed in independent MS experiments and/or by immunoblotting of selected proteins. For the corresponding GO analysis, the authors need to show that the GO results are different for the entire 1200 proteins compared to just IBT-sensitive for the results to be meaningful.

As suggested by the reviewer, we repeated the MS experiment in triplicate for each treatment to address the reproducible proteins that are IBT21-rescued aggregation-prone proteins (IBT-SENSITIVE proteins). The highly reproducible IBT-SENSITIVE proteins are shown in Figure 5—figure supplement 1. As expected, the molecular chaperone HSPA5 and the ER folding enzyme PDIA4 were observed in Figure 5—figure supplement 1A. In the case of the representative ER client proteins, we present transgelin-2 (TAGLN2) as a secreted protein and annexin A1 (ANXA1) as a membrane protein instead of cathepsin D (CTSD) and neutral α-glucosidase AB (GANAB) because CTSD and GANAB are synthesized at the ER but located in the lysosome. TAGLN2 and ANXA1 were also observed in Figure 5—figure supplement 1B. The reproducibility of the IBT-SENSITIVE proteins was supported by the GO enrichment analysis shown in Figure 5—figure supplement 1. The results of these experiments are shown in subsection “In vivo effects of IBT21 on protein aggregation during ER stress”, and in Figure 5—figure supplement 1B.

Regarding the GO-based comparison, GO enrichment results for the IBT-SENSITIVE proteins shown in Figure 5H-I and 5—figure supplement 1B-C were different from that of the IBT-bonded Competed and Selective proteins shown in Figure 6—figure supplement 1B-C. Nevertheless, the Competed and Selective proteins contained approximately 60% of the IBT-SENSITIVE proteins in each enriched GO group shown in Figure 6H, indicating that the Competed and Selective proteins largely cover the IBT21-targeted proteins corresponding to the chemical chaperone activity. The results of this experiment are shown in subsection “In vivoeffects of IBT21 on protein aggregation during ER stress”, and in Figure 5, Figure 5—figure supplement 1 and Figure 6.

8) In the MS experiment shown in Figure 6E that is measuring the ability of IBT22 to block labeling of proteins by the photocrosslinking probe, the data presentation makes it difficult to determine which targets are the most significantly blocked from labeling and by how many fold compared to DMSO. It would be helpful if the authors would adopt a more standard presentation of the data such as that shown in Lanning et al., 2014). The authors must clarify what threshold (fold-change) is being used to determine a protein is a bona fide target versus nonspecific target. Furthermore, how many total proteins from Figure 6E overlap with those in the IBT-sensitive fraction from Figure 5G?

We re-examined the IBT-BOUNDED proteins in a pie chart in the same style as reported in Lanning et al., (2014) (Figure 6E). The threshold (fold-change) is added to the text in subsection “In vivoeffects of IBT21 on protein aggregation during ER stress”, and is also shown in Figure 6D. The number of overlapped proteins with those in the IBT-SENSITIVE fraction from Figure 5G is shown in Figure 6F.

9) Finally, the authors must assess whether the IBT compound series functions similarly to ameliorate ER stress in other cell lines – preferably primary cells.

We successfully presented the same effects of IBT21 on another cell line, Hap1 cells. The results of this experiment are shown in subsection “Structure-activity relationship study of imidazo[2,1-*b*]benzothiazole derivatives (IBTs)”, and in Figure 3—figure supplement 1.

[Editors' note: further revisions were requested prior to acceptance, as described below.]Essential Revisions:1) New experiments are only included as supplementary figures. Selected key data, and instances in which experiments were repeated with more appropriate controls, should be incorporated into the main figures.

As suggested, we swapped Figure 4E and Figure 4-figure supplement 1.

2) In the Discussion section, please add a section addressing the surprising finding that the heat shock response is not influenced by their compounds.

We added to the Discussion section according to the reviewer’s comment.

3) There are 58 overlapped proteins for the Competed and Selective (1205 proteins) and IBT-sensitive (125 proteins) groups. However, only two are highlighted and they are not among the strongest hits from either group. Can anything be learned from overlapping stronger hits? Does this data suggest IBT has over 1000 off-targets (a lot) or is the conclusion that it is stabilizing many more proteins that are not detected in the experiment that determined IBT-sensitive compounds?

We agree with the latter of the reviewer's comments, and described in the Results section.

4) For the SAR discussion, it would be useful to highlight what is preferred and what is not preferred as opposed to just what is preferred.

Thank you for raising this point; we described what is preferred and what is not preferred in the Results section.